# The Protective Pathways Activated in Kidneys of αMUPA Transgenic Mice Following Ischemia\Reperfusion-Induced Acute Kidney Injury

**DOI:** 10.3390/cells12202497

**Published:** 2023-10-20

**Authors:** Heba Abd Alkhaleq, Tony Karram, Ahmad Fokra, Shadi Hamoud, Aviva Kabala, Zaid Abassi

**Affiliations:** 1Department of Physiology and Biophysics, Rappaport Faculty of Medicine, Technion-Israel Institute of Technology, Haifa 3109601, Israel; heba.abd.alkhaleq@gmail.com (H.A.A.); afokra@gmail.com (A.F.); avivak@technion.ac.il (A.K.); 2Department of Vascular Surgery, Rambam Health Care Campus, Haifa 3109601, Israel; t_karram@rambam.health.gov.il; 3Internal Medicine, Rambam Health Care Campus, Haifa 3109601, Israel; s_hamoud@rambam.health.gov.il; 4Laboratory Medicine, Rambam Health Care Campus, Haifa 3109601, Israel

**Keywords:** acute kidney injury, αMUPA, biomarkers, leptin, inflammation

## Abstract

Despite the high prevalence of acute kidney injury (AKI), the therapeutic approaches for AKI are disappointing. This deficiency stems from the poor understanding of the pathogenesis of AKI. Recent studies demonstrate that αMUPA, alpha murine urokinase-type plasminogen activator (uPA) transgenic mice, display a cardioprotective pathway following myocardial ischemia. We hypothesize that these mice also possess protective renal pathways. Male and female αMUPA mice and their wild type were subjected to 30 min of bilateral ischemic AKI. Blood samples and kidneys were harvested 48 h following AKI for biomarkers of kidney function, renal injury, inflammatory response, and intracellular pathways sensing or responding to AKI. αMUPA mice, especially females, exhibited attenuated renal damage in response to AKI, as was evident from lower SCr and BUN, normal renal histology, and attenuated expression of NGAL and KIM-1. Notably, αMUPA females did not show a significant change in renal inflammatory and fibrotic markers following AKI as compared with wild-type (WT) mice and αMUPA males. Moreover, αMUPA female mice exhibited the lowest levels of renal apoptotic and autophagy markers during normal conditions and following AKI. αMUPA mice, especially the females, showed remarkable expression of PGC1α and eNOS following AKI. Furthermore, MUPA mice showed a significant elevation in renal leptin expression before and following AKI. Pretreatment of αMUPA with leptin-neutralizing antibodies prior to AKI abolished their resistance to AKI. Collectively, the kidneys of αMUPA mice, especially those of females, are less susceptible to ischemic I/R injury compared to WT mice, and this is due to nephroprotective actions mediated by the upregulation of leptin, eNOS, ACE2, and PGC1α along with impaired inflammatory, fibrotic, and autophagy processes.

## 1. Introduction

Globally, acute kidney injury (AKI) occurs in 2–5% of hospitalized patients and has a significant impact on morbidity and healthcare services [1,2,3,4]. AKI is the most frequently observed syndrome in critically ill patients, affecting up to 20% of patients in intensive care units (ICUs) in the USA [5]. Mortality from AKI remains high; up to 60% of patients with severe AKI in the ICU die from the disease [6]. Between 6.1 and 22.4% of individuals undergoing major noncardiac surgery develop AKI [7]. AKI increases the risk of, and may lead to, chronic kidney disease (CKD) or end-stage renal disease, both of which affect other organs such as the heart [8]. Serum creatinine (SCr) or urine output (UO) are the factors used to define AKI stage [9]. However, sCr may not change until approximately 50% of kidney function has been lost [10]. Therefore, novel biomarkers for kidney injury have been introduced. These markers include neutrophil gelatinase-associated lipocalin (NGAL) and kidney injury molecule (KIM-1) [11,12,13].

Ischemia–reperfusion injury (I/R) is the primary pathophysiological process, accounting for 60% of the cases of AKI [14]. In addition to ischemic damage, the pathophysiological mechanisms of AKI also involve inflammatory cell migration, mitochondrial damage, oxidative stress, stimulation of the renin–angiotensin system, and reduction in nitric oxide (NO) [9]. Differences in the severity of AKI have been observed in various experimental models. For instance, studies have reported protective effects of the female sex on the severity of AKI following I/R [15,16,17], while, conversely, males exhibit a higher susceptibility to AKI compared to females [18,19]. This sexual dimorphism is attributed to sex hormones (estrogen and testosterone), which affect endothelial NO synthase (eNOS) and other mediators [20,21,22]. Several studies have suggested that estrogen may protect mice from AKI [16,23,24]. Moreover, the renin–angiotensin–aldosterone system (RAAS) is also influenced by sex hormones. In this regard, males exhibit greater RAAS activity than females and, thus, may benefit from the cardioprotective effects of ACE inhibition and angiotensin-receptor blockers [25,26]. Estrogen reduces renin levels, ACE activity, angiotensin II receptor type 1 density, aldosterone secretion, angiotensin II activity, and the downstream effects of angiotensin II [26,27]. Estrogen further inhibits the typical RAAS pathway, increasing the activity of the ACE2-Ang1-7 axis [25]. Differences in sex hormones between men and women also influence oxidative stress and other renal injury mechanisms independent of RAAS hormones. Studies have shown that androgens raise renal and systemic oxidative stress, whereas estradiol has antioxidant properties [28,29]. It has also been demonstrated that estradiol reduces ischemic kidney injury in rats by increasing NO bioavailability [30]. Support for NO’s nephronprotective role is derived from the observation that NO donors and eNOS activators ameliorate experimental renal I/R injury [31,32]. 

Another potential contributor to the pathogenesis of AKI is urokinase-type plasminogen activator (uPA) and its receptor (uPAR), although their roles are somewhat vague and, at times, controversial. The proximal and distal tubule’s epithelial cells generate a great amount of uPA, which is secreted into the tubular lumen [33]. uPA is a serine protease involved in tissue remodeling and cell migration [34,35]. The active uPA converts plasminogen to active plasmin, thus degrading intravascular fibrin and extracellular matrix (ECM) or activating latent growth factors such as transforming growth factor β1 (TGF- β1) [36]. Local renal uPAR expression plays a pivotal role in the pathogenesis of I/R AKI in a mouse model and in acute kidney allograft rejection. Specifically, uPAR deficiency protects kidney tissue from the generation of ROS and, consequently, from severe apoptosis in I/R injury and acute kidney allograft rejection [37]. Furthermore, the upregulation of the uPA/uPAR system has been observed in vitro under conditions of hypoxia/reperfusion [38,39]. To understand the involvement of uPA/uPAR in the pathogenesis of AKI, we utilized alpha murine urokinase-type plasminogen activator (αMUPA) transgenic mice whose phenotype is similar to that of caloric restricted (CR) animals and who spontaneously consumed 20% less food and lived 20% longer than their wild-type (WT) counterparts. Body weight, body length, and blood sugar of αMUPA mice were also decreased. Additionally, αMUPA mice retained a youthful appearance even as they became older [40]. αMUPA mice were originally created as a tool to study the potential involvement of uPA in eye pathophysiology [41,42]. However, analysis of the body organs revealed that uPA is also overexpressed in the brain, which affects appetite, food intake, and leptin levels. Subsequently, this may affect the behavior of other organs including the kidney [43,44]. The changes seen in αMUPA mice are activated by uPA overexpression in the trigeminal nucleus in the brain. The trigeminal nucleus is connected to the dorsal motor of the vagus nerve and nucleus tractus solitarius; two regions of the brain that control appetite. This connection may help innervate adipose tissue through the autonomic nervous system, leading to an increase in the secretion of the adipokine hormone leptin. Leptin is a 16 Kd adipokine hormone produced by adipocytes which controls food intake and energy consumption [45,46]. Due to elevated leptin levels in the serum, αMUPA mice have a stable weight and are resistant to obesity. Pro-opiomelanocortin (POMC) and cocaine and amphetamine-regulated transcript (CART) are expressed by anorexigenic neurons in the hypothalamus arcuate nucleus in response to leptin, a satiety signal produced by adipose tissue [47,48]. uPA mRNA has also been detected in neuronal cells of the hypothalamus paraventricular nucleus (PVN), an area linked to eating behavior, in the brain of αMUPA mice. uPA mRNA is not normally found at this site [43].

Interestingly, the αMUPA female mice that were subjected to myocardial infarction (MI) showed increased fractional shortening, decreased infarction duration, inflammatory response, cardiac ageing, and elevated serum leptin levels [41]. The latter plays a cardioprotective role, as is evident by the reduced fractional shortening and post-MI infarction size in the αMUPA when leptin-neutralizing antibodies were administered [44]. Furthermore, another study demonstrated that the expression of TNF-α (tumor necrosis factor alpha) and IL-1β (Interleukin-1 beta) encoding proinflammatory genes were increased in the wild-type (WT) heart after MI and following LPS (lipopolysaccharide) administration. These genes were expressed to a much lower degree in αMUPA mice [49,50]. Remarkably, when these animals were pretreated with leptin-neutralizing antibodies, their resistance to MI or LPS damage was abolished. Based on these findings, we hypothesize that αMUPA mice may exhibit resistance to AKI, akin to their resilience to MI. Therefore, the current study aimed to investigate if αMUPA transgenic mice have a protective leptin-mediated pathway activated in the kidneys or circulation following I\R-induced AKI. We particularly focused on comparing the responses of females to their WT counterparts and male gender. This notion is particularly supported by the findings that αMUPA female mice exhibit higher levels of circulating leptin than their WT animals [44,51].

## 2. Materials and Methods

### 2.1. Animals

WT FVB/N and transgenic αMUPA mice were obtained from Prof. Ruth Miskin, Weizmann Institute of Science (Rehovot, Israel), and were bred at the Rappaport Faculty of Medicine (Haifa, Israel). Mice were housed five per cage, in a facility with a controlled temperature room, and fed standard mice chow. All experiments were carried out in accordance with the guidelines of the committee for the supervision of animal experiments, Technion, IIT (IL 097-06-19).

Induction of AKI: WT and αMUPA male (*n* = 5–12) and female (*n* = 5–6) mice (males 22–35 g, females 17–26 g) of the same age (12 weeks) were anesthetized with a mixture of 100 mg/kg ketamine and 10 mg/kg xylazine. Mice were dorsally placed on a controlled thermos-regulated table to maintain their core body temperature at 37 °C. The abdominal wall was opened, and the intestines were covered with saline-soaked gauze to minimize dehydration. The renal arteries of the mice in the experimental groups were exposed and clamped bilaterally by using vascular clamps for 30 min to produce ischemia, followed by reperfusion. After reperfusion, blood flow was reestablished with visual verification, the abdomen was sutured, and the animal returned to its cage until sacrifice. The reperfusion time determined from removing the vascular clamps, and its impact on SCr and BUN levels, histological sections, and the expression of relevant biomarkers was examined after 48 h. Sham-operated animals served as controls and underwent the same surgical procedure as the experimental groups but without the renal artery clamping.

Pretreatment with leptin-neutralizing antibodies: An additional group (*n* = 4) consisting of both αMUPA males and females underwent AKI induction in the presence of antileptin. Specifically, the AF489 leptin-neutralizing antibody (R&D Systems Inc., Minneapolis, MN, USA) was injected (5 μg/mouse) through the tail vein for two consecutive days before AKI induction (the last injection was two hours before AKI surgery). 

### 2.2. Evaluation of Blood Variables and Renal Functional Parameters

Forty-eight hours following the AKI procedure, mice were anesthetized with a mixture of 100 mg/kg ketamine and 10 mg/kg xylazine. Blood samples (500 μL each) were drawn through the left ventricle of the heart and centrifuged at 1500 rpm for 10 min at 4 °C. SCr and blood urea nitrogen (BUN) levels were determined in mice serum using commercial kits (Siemens, Germany) with an autoanalyzer dedicated instrument (Dimension RXL, Siemens, Germany).

### 2.3. Histopathology

Kidneys were removed from all experimental groups, fixed in 4% formalin, and embedded in paraffin. Transverse sections were stained with hematoxylin and eosin (H&E). Briefly, paraffin tissue sections were rehydrated under standard protocols, including clearing in xylene (3 times), and rehydrated in anhydrous alcohol (100% alcohol and 95% alcohol) and distilled water, respectively. Paraffin tissue sections were stained with hematoxylin for 2 min, rinsed using tap water, and blotted dry. Next, slides were incubated with eosin stain for 2 min and followed by routine dehydration, including 95% alcohol and xylene. Finally, slides were sealed with a slide-mounting medium, DPX. Histological analysis of the renal tissue for casts, necrosis, and inflammation, for both WT mice and αMUPA animals subjected either to sham operation or AKI, was performed.

### 2.4. Real-Time PCR

Complete RNA was purified from frozen kidney samples (cortex and medulla) using TRIzol reagent. cDNA was synthesized according to the manufacturer’s protocol from complete RNA using the maximal first strand cDNA synthesis kit for RT-qPCR. Using PerfeCTa SYBR Green with the target gene primers, quantitative real-time PCR analysis was performed and analyzed in the 7500 Real Time PCR System (Applied Biosystems, RHENIUM 8440, Foster City, CA, USA). The mRNA levels of the various genes (uPA, uPAR, eNOS, ACE2, IL-6, leptin, LC3, KIM-1, NGAL and peroxisome proliferator-activated receptor-gamma coactivator (PGC1α)) were standardized to mRNA levels of Rpl13a, referred to as the housekeeping gene. Relative to the normalized values obtained for WT mice at baseline, fold shift was measured.

Mouse primers:uPA: -F-AGAGTCTGAAAGTGACTATCTC,-R-CCTTCGATGTTACAGATAAGCuPAR: -F-TCTGGATCTTCAGAGCTTTC,-R-GCCTCTTACGGTATAACTCCPAI-1: -F-AGCAACAAGTTCAACTACAC,-R-CTTCCATTGTCTGATGAGTTCInsR: -F-AAGACCTTGGTTACCTTCTC,-R-GGATTAGTGGCATCTGTTTGeNOS: -F-AAAGCTGCAGGTATTTGATG,-R-AGATTGCCTCTATTTGTTGCACE2: -F-CATTTGCTTGGTGATATGTG,-R-GCCTCTTGAAATATCCTTTCTGMasR: -F-GTTTAAGGAACTCTGGAAGATG,-R-TTAGTCAGTTAGTCAGTGGCRenin: -F-AGCCAAGGAGAAGAGAATAG,-R-CTCCTGTTGGGATACTGTAGIL-6: -F-GTCTATACCACTTCACAAGTC,-R-TGCATCATCGTTGTTCATACIL-10: -F-CAGGACTTTAAGGGTTACTTG,-R-ATTTTCACAGGGGAGAAATCTLR4: -F-TCCCTGCATAGAGGTAGTTCC,-R-TCCAGCCACTGAAGTTCTGALeptin: -F-ACATTTCACACACGCAGTCGG,-R-GGACCTGTTGATAGACTGCCALC3: -F-GAACCGCAGACGCATCTCT,-R-TGATCACCGGGATCTTACTGGP62: -F-AATGTGATCTGTGATGGTTG,-R-GAGAGAAGCTATCAGAGAGGGalectine 8: -F-ATATACAAAAGCCAGGCAAG,-R-CAAATGCTTTCACATTGAGGTGF-β: -F-GGATACCAACTATTGCTTCAG,-R-TGTCCAGGCTCCAAATATAGCaspase 3: -F-CATAAGAGCACTGGAATGTC,-R-GCTCCTTTTGCTATGATCTTCCaspase 7: -F-CAAAACCCTGTTAGAGAAACC,-R-CCATGAGTAATAACCTGGAACKIM-1: -F-CTGGAGTAATCACACTGAAG,-R-AAGTATGTACCTGGTGATAGCNGAL: -F-ATATGCACAGGTATCCTCAG,-R-GAAACGTTCCTTCAGTTCAGPGC1α: -F-TCCTCTTCAAGATCCTGTTAC,-R-CACATACAAGGGAGAATTGCRpl13a: -F-AAGCAGGTACTTCTGGGCCG,-R-GGGGTTGGTATTCATCCGCT

### 2.5. Western Blot

Samples of kidney tissue (20 mg), including cortex and medulla, were homogenized in a lysis buffer and protein was quantified using Bradford commercial assay. Protein samples (50 μg) were electrophoresed on sodium dodecyl sulfate (SDS) polyacrylamide gel (10 percent) under denaturing conditions, and then electrotransferred to nitrocellulose membranes for 1.5 h at 100 V. Membranes were blocked with 5 percent BSA in Tris-buffered saline (TBS) for 1 h at room temperature. Main antibodies were used in TBST at concentrations of 1:200–1:1000 with 5% BSA overnight at 4 °C. In order to serve as internal control, immunodetection of GAPDH with monoclonal anti-GAPDH antibodies was tested. HRP-secondary antibodies were applied for 45 min at room temperature at a concentration of 1:15,000. The signal was detected with ECL (chemiluminescence substrate), and images were captured with ImageQuant LAS 4000.

### 2.6. Quantitative Inflammation Array

Serums from sham-operated mice and AKI-operated mice were collected. The concentration of various chemokines and cytokines in serum was determined using the Quantibody Mouse Inflammation Array Kit according to the manufacturer’s instructions (RayBiotech, Inc., Norcross, GA, USA). 

### 2.7. Statistical Analysis

Animals were randomly assigned to the experimental group. The results are presented as mean ± SEM. -Statistical significance was tested for comparisons between WT and αMUPA mice using unpaired Student’s t-tests. A *p* < 0.05 value was found to be statistically significant.

## 3. Results: Impact of I/R-Induced AKI on

### 3.1. Body and Kidney Weight

αMUPA mice, especially females, have a lower basal body weight (BW) compared with their WT counterparts (Table 1). While AKI had no impact on the body weight of αMUPA female mice compared to their counterparts subjected to sham procedures, it did result in a slight decrease in male mice. In contrast, in the WT subgroup, body weight was significantly reduced following AKI in male mice and, to a lesser extent, in female mice, compared to their respective sham-operated counterparts. Kidney weight (KW) increased following AKI as compared to their sham counterparts in both WT and αMUPA mice (Table 1). However, αMUPA mice, especially females, showed only a slight change in KW after AKI compared to the significant increase in KW observed in their WT counterparts subjected to a similar procedure (Table 1). These differences persisted even after normalization of KW to BW (Table 1). To evaluate whether leptin plays a protective role against AKI in αMUPA mice, I.V. leptin-neutralizing antibody (AF498) was administered to these animals prior to AKI induction. Pretreatment with AF498 led to a significant decrease in BW in αMUPA female mice following AKI, with a milder effect observed in male animals (Table 1). KW increased in both male (0.96 ± 0.05 as compared to 0.79 ± 0.02, *p* < 0.05) and female (0.82 ± 0.09 as compared 0.59 ± 0.02, *p* < 0.005) following the administration of AF498 to αMUPA mice prior to AKI operation. These results show that leptin neutralization reverses post-ischemic αMUPA renal protection.

### 3.2. Kidney Function and Renal Injury Biomarkers

Induction of AKI increased sCr 48 h after injury in both WT males (from 0.3 ± 0.01 to 2.0 ± 0.36, *p* = 0.001) and αMUPA males (from 0.32 ± 0.01 to 1.07 ± 0.28, *p* = 0.1), yet the latter subgroup exhibited a markedly reduced increase in sCr as compared to WT (Figure 1A). Similarly, the elevation of BUN following AKI was significantly reduced in αMUPA males compared to their WT counterparts (Figure 1B). However, sCr and BUN levels in females of both subgroups were not significantly affected by I/R-AKI. Additionally, sCr and BUN levels in αMUPA females and males were not significantly affected by AF498 pretreatment (Figure 1E,F). Noteworthy, WT mice (males and females) and αMUPA male mice displayed significant increases in renal expression of both NGAL and KIM-1, biological markers of AKI, following AKI (Figure 1C,D). WT male mice were the most susceptible subgroup to AKI, as was evident by profound elevation in NGAL expression (~250-fold) (Figure 1C). While renal expression of NGAL and KIM-1 in αMUPA females did not change following AKI, these biomarkers increased in WT females that underwent similar renal I/R AKI. These results demonstrate that αMUPA, mice especially females, exhibit an increased tolerance to ischemic renal stress, as was evident by resistance to I/R injury 48 h following AKI. Of note, renal expression of NGAL and KIM-1 in αMUPA females, which did not change following AKI as described in Figure 1C,D, was significantly augmented by AF498 administration (*p* < 0.005) (Figure 1G,H). 

### 3.3. Renal Histology

Figure 2 depicts the renal histological alterations in WT and αMUPA mice subjected to I/R-AKI. A distinct pattern in histological renal response was observed, both visually and microscopically, in WT and αMUPA kidneys following AKI. This evidence suggests a protective role of uPA against renal ischemia (Figure 2). Specifically, WT mice exhibited more exaggerated injury to kidney tissue, including tubular collapse, loss of the brush border, and cellular detachment from tubular basement membranes (Figure 2A). Necrosis and the presence of hyaline casts were observed in the outer medulla, whereas congestion was more intense in the inner medulla. In contrast, αMUPA mice, especially the females, exhibited attenuated kidney injury in response to I/R-AKI and showed similarity to sham kidney tissues (Figure 2B). The observed resistance of αMUPA to I/R-AKI was abolished when these animals were pretreated with neutralizing leptin Abs. Exaggerated renal injury was exhibited, as was evident by tubular collapse, loss of the brush border, and cellular detachment from tubular basement membranes (right column of Figure 2A,B). 

### 3.4. Renal uPA and uPA Receptor 

WT mice (males and females) exhibited a reduction in renal uPA expression/abundance (Figure 3A,D) along with a significant elevation in uPA receptor expression (~12-fold) following AKI (Figure 3B). αMUPA males displayed enhanced uPA receptor mRNA expression following AKI, but to a lesser extent than WT mice (Figure 3B). In contrast, renal uPA and uPA receptor expression in αMUPA females was not changed following AKI (Figure 3A,B). It is important to note that female mice from both subgroups of mice have a lower basal uPAR immunoreactivity as compared with male mice (Figure 3E). 

A specific glycoprotein, plasminogen activator inhibitor 1 (PAI-1), binds to two-chain of uPA and blocks the proteolytic action of uPAR-bound uPA [52]. Analysis of renal PAI-1 expression revealed that its expression was increased in WT subgroups following AKI, and more profoundly in male mice (~80-fold) (Figure 3C). In contrast, αMUPA female mice did not exhibit significant changes in renal PAI-1 expression, whereas a remarkable enhancement of PAI-1 expression was observed following AKI in WT mice. When αMUPA female mice were subjected to AKI in the presence of AF498, a significant reduction in renal uPA expression/abundance (Figure 3F,I) along with a significant elevation in uPA receptor expression (~2.5-fold) and abundance (~5-fold) (Figure 3G,J) were observed.

In contrast, renal uPA and uPA receptor immunoreactivity in αMUPA males were not affected by AF498 administration (Figure 3I,J). Noteworthily, uPA expression was reduced after AF498 injection in αMUPA males (Figure 3F). Renal PAI-1 expression was decreased in both subgroups following AF498 pretreatment (Figure 3H).

### 3.5. Renal Leptin, Insulin Receptor, and PGC1α

Figure 4 depicts the expression of three key modulators of kidney function in the various experimental groups. The insulin receptor (InsR) is crucial for glomerular and tubular function [53]. When insulin signaling is diminished, as may occur in insulin-resistant states, it may lead to a number of substantial renal complications, including albuminuric glomerular disease and hypertension [53]. αMUPA mice (males and females) showed a significant elevation in renal leptin expression both before and following AKI, as compared to their comparable WT mice (Figure 4A). WT mice, both males and females, showed a significant reduction in renal InsR expression following AKI; yet, WT male mice displayed a more profound reduction than females (Figure 4B). In contrast, αMUPA male mice exhibited an attenuated reduction in renal InsR as compared to male WT animals. Noteworthily, renal InsR expression in αMUPA female mice did not change following AKI and exhibited comparable levels to those of healthy sham control. 

The expression of PGC-1α in the kidneys of WT male and female mice and in αMUPA male mice was decreased following AKI (Figure 4). However, a sharper decrease in PGC-1α was obtained in WT male mice compared to all other subgroups (Figure 4C). αMUPA female mice did not show a change in renal PGC-1α expression following AKI. Noteworthily, αMUPA female mice had the highest expression level of PGC-1α among the four studied subgroups, both at baseline condition and following AKI. Considering the beneficial role of PGC-1α as antifibrotic modulator, these results suggest that the high levels of renal PGC-1α in αMUPA female mice kidney may be involved in the attenuated susceptibility of these mice to characteristic ischemic injury in AKI. αMUPA mice (males and females) administered with AF498 prior to AKI exhibited a significant reduction in renal leptin (Figure 4D), InsR (Figure 4E), and PGC1α (Figure 4F) compared to αMUPA mice without AF498 injection. 

Collectively, αMUPA female mice exhibited elevated baseline levels of the beneficial parameters InsR and PGC-1α. Inhibition of leptin in these animals amplified their susceptibility to characteristic ischemic injury in AKI. 

### 3.6. Renal Expression of Inflammatory and Fibrotic Markers

WT mice (males and females) and αMUPA male mice showed a significant increase in renal Interleukin 6 (IL-6) expression, a proinflammatory cytokine, 48 h following AKI (Figure 5A). Noteworthy, male WT mice displayed the sharpest elevation in IL-6 (Figure 5A). In contrast, αMUPA females did not exhibit a significant change in renal IL-6 following AKI. Analysis of renal IL-6 immunoreactive levels showed that αMUPA male mice had a reduced amount of IL-6 protein compared to WT male mice following AKI. Additionally, there was no difference between the females of both WT and αMUPA subgroups; both subgroups exhibited reduction in IL-6 following AKI (Figure 5D). Toll like receptor 4 (TLR-4) expression remained stable in the experimental groups, with the exception of αMUPA female mice 48 hours after AKI, where it exhibited a notable decrease (Figure 5B). Interleukin 10 (IL-10), an anti-inflammatory cytokine, was unchanged in WT male mice 48 h after AKI, but increased in WT female mice and αMUPA male mice (Figure 5C). Notably, αMUPA females did not exhibit IL-10 expression in their renal tissue at baseline (Figure 5C). Analysis of renal immunoreactive levels of signal transducer and activator of transcription 3 (STAT3), p-STAT3, CathepsinL, I Kappa B (IKB), and mitogen-activated protein kinase (MAPK), a proinflammatory marker, revealed elevated levels in the renal tissue of WT mice (both males and females) following AKI (Figure 5E–I). In contrast, αMUPA mice, especially the females, exhibited a subdued increase in these markers; however, the expression of proinflammatory markers in this group was not affected by AKI (Figure 5E–I). AKI induction provoked renal transforming growth factor β (TGFβ) expression in WT (males and females) and males of αMUPA mice (Figure 5J). The increase observed in WT males was more remarkable than their αMUPA controls. Of interest, αMUPA female mice did not exhibit changes in renal TGFβ expression and immunoreactive levels following AKI induction (Figure 5J,K). αMUPA mice (males and females) that were subjected to AKI did not exhibit a significant change in renal IL-6 following AF498 pretreatment as compared to αMUPA mice without AF498 injection (Figure 5L). TLR-4 expression decreased after AF498 pretreatment in both male and female mice; however, αMUPA female mice displayed significantly higher levels of TLR4 expression compared with αMUPA males (Figure 5M). Renal immunoreactive expression of STAT3 and p-STAT3, proinflammatory markers, of αMUPA mice (males and females) were elevated following AF498 injection. Specifically, αMUPA female mice that exhibited a minimal elevations of STAT3 and p-STAT3 following AKI in the absence of AF498 (Figure 5E,F) displayed significant increase in these markers when pretreated with AF498 injection, showing a pattern similar to that of αMUPA males (Figure 5N,O). These results demonstrated that αMUPA mice, especially females, lost their resistance to AKI at the inflammatory level after inhibiting leptin by AF498. AF498 treatment prior to AKI increased renal IKB and MAPK abundance in αMUPA female mice to comparable levels of those obtained in αMUPA male animals (Figure 5P,Q). IL-10 was unchanged in αMUPA male mice that were subjected to AKI in the presence of AF498, but increased in αMUPA female mice (Figure 5R). Interestingly, renal expression of αMUPA females that did not express IL-10 in the absence of AF498 showed an expression of this anti-inflammatory cytokine in their kidneys following AF498 pretreatment, reaching levels comparable to those in αMUPA males.

Analysis of renal TGFβ expression in αMUPA after AF498 revealed an increase in females that was more remarkable than their male counterparts (Figure 5S). These results demonstrate that leptin is beneficial to the nephroprotective actions against AKI-induced fibrosis in αMUPA mice, especially females.

### 3.7. Renal Apoptotic and Autophagy Markers

Renal expression of Caspase 3 and 7 increased following AKI in both male and female WT mice and is presented in Figure 6. αMUPA male mice displayed an attenuated renal expression of Caspase 7 following AKI as compared with their WT controls. Kidney expression of these parameters did not change in αMUPA female mice following AKI (Figure 6A,B). The renal expression of various autophagy markers, including the microtubule-associated protein 1A/1B-light chain 3 (LC3), P62, and Galectine 8, were remarkably low in αMUPA female mice both during normal conditions and following AKI (Figure 6C–F).

αMUPA mice, both females and males, subjected to AKI exhibited a significant increase in Caspase 3 following AF498 pretreatment compared to untreated αMUPA mice (Figure 6G). Caspase 7 expression increased significantly following AF498 administration in both male and female αMUPA mice (Figure 6H). αMUPA females, but not males, exhibited elevated renal expression of autophagy markers, including LC3, P62, and Galectine 8, following AF498 injection compared to AKI-operated untreated αMUPA females (Figure 6I–L).

### 3.8. Renal Angiotensin Converting Enzyme 2 (ACE2) and MasR

While WT male mice showed a significant decrease in renal ACE2 and MAS receptor (MasR) expression/immunoreactivity following AKI (Figure 7A,B,D), WT females, αMUPA males, and αMUPA females did not, suggesting ACE2’s role in αMUPA kidney protection post-AKI. Furthermore, renin, a key enzyme in the RAAS known to contribute to hypertension and CKD [54,55], was lower in αMUPA females and remained unaffected by AKI in this subgroup (Figure 7C). αMUPA mice, both males and females, exhibited significant reductions in renal ACE2 and MasR expression, as well as ACE2 immunoreactivity, following AF498 injection prior to AKI (Figure 7E,F,H). The expression of renin significantly increased in αMUPA females and was further augmented by AF498 administration (Figure 7G).

### 3.9. Renal Endothelial Nitric Oxide Synthase (*eNOS)*

αMUPA female mice displayed higher levels of renal eNOS mRNA at baseline compared with WT mice and αMUPA males. This trend continued even after AKI (Figure 8C). WT male mice, which are vulnerable to ischemic insult, showed preserved eNOS immunoreactivity (while elevated in the other subgroups) and an increase in eNOS expression 48 h following AKI (Figure 8A,C), suggesting that eNOS plays a protective role in the kidneys of αMUPA female mice against ischemic injury. Additionally, WT male mice showed a significant reduction in p-eNOS following AKI. The other experimental groups did not exhibit such variation in p-eNOS immunoreactive protein levels before and after AKI (Figure 8B). αMUPA mice (males and females) exhibited increased renal eNOS and p-eNOS immunoreactive protein levels when pretreated with AF498 prior to AKI, in comparison to animals without AF498 administration (Figure 8D,E). Of notice, αMUPA mice (males and females) showed a reduction in eNOS expression following AKI in the presence of AF498 injection (Figure 8F), probably as a negative feedback to high protein levels of eNOS. 

In line with these findings, inhibiting leptin has a detrimental effect on the renal eNOS system, which otherwise has a beneficial effect on kidney function/histology during AKI.

### 3.10. Plasma Levels of Cytokines of WT and αMUPA Mice

The circulatory levels of proinflammatory chemokines, eotaxin, RANTES, lipopolysaccharide-induced CXC chemokine (LIX), and the tricarboxylic acid (TCA) were all elevated following AKI in WT male mice. In contrast, the concentrations of these chemokines did not change in αMUPA (males and females) and WT female mice. In actuality, αMUPA female mice exhibited the lowest levels of proinflammatory chemokines (Figure 9A–C). The growth factor granulocyte colony-stimulating factor (G-CSF), which affects the bone marrow to produce more white blood cells after inflammation or infection, increased in WT male mice but not in the other experimental subgroups (Figure 9E). Intercellular Adhesion Molecule 1 (ICAM-1), an essential receptor for leukocytes adhesion during inflammation, was elevated (~600-fold) in WT male mice compared with a negligible increase in αMUPA female mice (Figure 9F). While serum levels of TIMP metallopeptidase inhibitor 1 (TIMP-1) and IL-6, proinflammatory cytokines, increased approximately 2.5-fold in WT male mice 48 hours after AKI, the other subgroups exhibited attenuated levels (Figure 9G,H). The shifts in tumor necrosis factor *α* (TNF-α) receptors I and II levels showed a similar pattern (Figure 9I,J). Serum IL-10, an anti-inflammatory cytokine, decreased in WT (~5.5-fold) and in αMUPA male mice (~1.5-fold) who were subjected to AKI. αMUPA female mice exhibited higher IL-10 levels than WT female mice before and after AKI. Of notice, αMUPA female mice displayed the highest IL-10 levels in the serum (Figure 9K). One prominent characteristic of αMUPA mice is their high basal levels of the adipokine hormone leptin compared to WT mice. Furthermore, leptin levels were elevated in αMUPA female mice following AKI, but decreased in αMUPA males and WT female mice. Remarkably, the levels of leptin increased ~4811-fold following AKI in WT male mice (Figure 9L).

## 4. Discussion

The occurrence of AKI globally is increasing. Unfortunately, until now, the mechanisms underlying AKI have not been fully understood. The availability of αMUPA mice provides the most appropriate model to examine the involvement of uPA in the pathogenesis of AKI. We can study whether or not these animals exhibit resistance to ischemic renal injury as they did at the cardiac level [44,49]. In line with our assumption, the current study demonstrated that αMUPA mice, especially females, exhibited attenuated renal damage in response to AKI induction. This was evidenced by lower SCr, BUN, attenuated expression of kidney NGAL and KIM-1 (sensitive biomarkers for kidney injury), and normal renal histology compared to both male models and even WT females. As reported previously, αMUPA mice exhibit several phenotypic changes, including spontaneous reduction in food consumption, growth, and BW retardation, along with increased longevity [56]. In line with these characteristics, the present study demonstrated that αMUPA mice, especially females, have a lower BW compared with their WT. Interestingly, αMUPA female mice did not experience changes in BW following AKI, whereas both male and female WT subgroups showed significant reductions in BW after AKI. Similarly, KW increased following AKI in WT, but to a lesser extent in αMUPA mice, especially females. These differences suggest that renal edema, a hallmark of AKI, is less severe in αMUPA mice compared to WT mice. This reduced edema may have contributed to the milder elevation in biomarkers of kidney dysfunction (sCr and BUN) and renal injury (NGAL and KIM-1) observed in the αMUPA strain. In addition to their resistance to I/R-induced renal injury, αMUPA female mice did not exhibit significant increases in renal inflammatory and fibrotic markers, including IL-6, TLR4, and TGF-β, as well as circulatory proinflammatory cytokine levels following AKI when compared to WT mice and αMUPA males. These results are in concurrence with the reported decreased inflammatory response to LPS administration into αMUPA female mice [50]. Furthermore, another study demonstrated that the expression of proinflammatory TNF-α and IL-1β genes were increased in the WT hearts after MI, but to a lower degree in αMUPA mice [49]. These results demonstrate a lower inflammatory response to AKI and MI in αMUPA kidneys and hearts, especially in females. Likewise, αMUPA female mice exhibited the lowest levels of renal apoptotic and autophagy markers (LC3 and P62), under normal conditions and following AKI. Of note, αMUPA mice, especially females, exhibited high baseline levels of PGC1α and eNOS, both before and after AKI induction. Concomitantly, αMUPA mice exhibited preserved renal ACE2 immunoreactivity post AKI, suggesting a potential involvement of PGC1α, eNOS, and ACE2 in nephroprotection against AKI. The current finding that αMUPA females displayed the highest basal renal levels PGC-1α, and the fact that this subgroup did not display decline in PGC-1α following AKI compared with WT and αMUPA males, suggest a beneficial role of this mediator in the observed nephroprotection phenomenon. This notion is further supported by the fact that PGC-1α regulates various processes such as adipogenesis, angiogenesis, gluconeogenesis, heme biosynthesis, thermogenesis, cellular degeneration protection, and mitochondrial biogenesis [57,58]. Moreover, subjects deficient in PGC-1α experience a negative impact during AKI, gradually developing renal fibrosis and CKD [58]. The increased biogenesis of mitochondria through PGC-1 alpha overexpression promotes recovery from kidney injury in ischemic conditions [59,60]. Another study found that PGC-1α plays an important role in cellular recovery following an injury model of I/R [61]. Furthermore, PGC-1 knockout mice failed to recover following AKI. Collectively, these results suggest that the elevated renal PGC-1α levels in αMUPA female mice, both at baseline and following AKI, may contribute to their reduced susceptibility to ischemic injury in AKI, given PGC-1α’s known antifibrotic role. 

An additional hallmark of αMUPA mice, particularly females, is elevated circulating leptin relative to their wild-type (WT) FVB/N counterparts [43,44,56,62,63]. We extended these findings by demonstrating that αMUPA mice (males and females) showed a significant elevation in renal leptin expression before and following AKI. The exact mechanisms underlying the increased expression of renal leptin is unknown. However, stressful stimuli such as ischemic and inflammatory responses may provoke local leptin production in various organs besides the adipose tissue. This notion is supported by previous studies that showed enhanced production of leptin in cardiac tissue in response to ischemic [49] and inflammatory [50] stress. While adipose tissue is the primary source of leptin in the blood, we cannot rule out the possibility that the kidney’s upregulation of leptin production may contribute to circulating leptin concentrations. It should be emphasized that renal sympathetic nerves innervate the vasculature and nephrons, where they play a central role in the regulation of renal hemodynamics and tubular function, and probably affect local leptin expression [64,65]. 

Since previous studies have demonstrated that endogenous leptin plays a causal function in reducing dysfunction in fractional shortening and post-MI infarction size in the αMUPA heart by using leptin-neutralizing antibodies [44], we applied the same approach to explore the possibility that leptin is involved in the renoprotection of αMUPA females against AKI. Indeed, pretreatment of αMUPA mice with AF498 (leptin-neutralizing antibody) prior to AKI abolished their resistant to renal damage, as evidenced by increasing renal NGAL and KIM-1 expression, inflammatory and fibrotic response, reducing renal eNOS, ACE2, and PGC-1α along activation of renin. Furthermore, pretreatment with AF498 aggravated the histological renal aberrations characterizing AKI, as was evidenced by tubular collapse, loss of the brush border, and cellular detachment from tubular basement membranes, especially in αMUPA females, which were resistant to AKI in the absence of leptin antagonist. These findings could be attributed to the previously reported antiapoptotic effect of leptin [66,67,68]. Our findings, which show increased renal expression of leptin following AKI in αMUPA mice, may explain the observed reduction in the expression of proinflammatory genes in these mice. While treated αMUPA female animals did not exhibit significant changes in renal inflammatory and fibrotic markers such as IL-6, STAT3, p-STAT3, and TGF-β following AKI, in contrast to their male counterparts and WT mice, these proinflammatory and fibrotic markers showed elevation after AF498 injection in αMUPA female mice. Additional support for the anti-inflammatory impact of leptin comes from the observation that αMUPA mice that were subject to MI displayed lower levels cardiac proinflammatory genes, TNFα and IL-1β, compared to a significant increase in those parameters in WT mice subjected to the same cardiac insult [49]. Furthermore, in vivo studies revealed that pretreatment with exogenous leptin largely reduced the mRNA levels of cultured cardiomyocytes TNFα and IL-1β exposed to hypoxia [49]. The nephroprotective mechanisms of leptin in αMUPA female mice can be attributed to their high renal levels of beneficial parameters, including InsR and PGC-1α, in the absence of AF498. Conversely, these parameters decreased in the presence of leptin-neutralizing antibodies. These findings demonstrated that αMUPA mice, especially females, lost their resistance to AKI at the inflammatory and fibrotic level after the inhibition of leptin by AF498. 

αMUPA female mice exhibited the lowest levels of renal apoptotic (Caspase 3 and 7) and autophagy markers (LC3 and P62) during normal conditions and following AKI in the absence of AF498. The injection of AF498 exaggerated the apoptotic and autophagy markers in αMUPA mice, particularly in females. This supports the idea that leptin plays a beneficial role in reducing apoptosis and autophagy following AKI in αMUPA mice. ACE2, eNOS, and estrogen may be also involved in the protective pathway of αMUPA female kidneys following AKI. αMUPA males and females did not show a significant decrease in the expression of ACE2 following AKI without AF498 pretreatment; however, ACE2 was reduced significantly after AF498 I.V. injection. These findings demonstrate that leptin affects the levels of ACE2 in the kidneys of αMUPA mice. The inhibition of leptin also affected the renal eNOS system, which has a beneficial effect on kidney function during AKI [69]. Prior to AF498 administration, female mice exhibited elevated renal eNOS expression both at baseline and after AKI compared to WT animals. WT male mice showed a massive reduction in p-eNOS following AKI; the other experimental groups did not exhibit such a decrease in p-eNOS immunoreactive protein levels before and after AKI. Following AF498, e-NOS mRNA was reduced significantly in αMUPA males and females and e-NOS and p-eNOS protein levels were elevated significantly. 

uPA exerts a wide range of biological actions, including cell migration, matrix degradation by activating prometalloproteinases (pro-MMPs), and cell signaling by releasing active growth factors, such as VEGF and core fibroblast growth factor (βFGF), from the matrix [34,35,70]. Moreover, active uPA converts plasminogen (the inactive zymogen) into active plasmin, thereby degrading intravascular fibrin and extracellular matrix (ECM), or modulating transforming growth factor β1 (TGF- β1) [36]. The latter was shown to be upregulated in WT mice that were subjected to AKI, but not in αMUPA transgenic animals. Interestingly, αMUPA female mice did not show enhancement in renal TGFβ expression and immunoreactive levels following AKI induction. This observation may explain their attenuated histological alterations. In addition, uPA degradation is a multistep process involving other molecules. A specific enzyme named plasminogen activator inhibitor 1 (PAI-1) binds to two chains of uPA, thereby blocking the proteolytic action of uPAR-bound uPA [52]. Analysis of renal PAI-1 revealed that its expression was increased in WT subgroups following AKI, and even more strongly in male mice. In contrast, αMUPA female mice did not display significant changes in the expression of renal PAI-1, as compared to the remarkable enhancement in PAI-1 levels observed in WT mice following AKI. This unique behavior of αMUPA females, namely, lower levels of PAI-1, may unleash uPA, thus potentiating its renal effects. Moreover, upregulation of the uPA/uPAR system has been demonstrated under conditions of hypoxia/reperfusion in vitro [38,39]. In addition, uPA/uPAR activation has been correlated with allograft rejection in a human biopsy study [71,72]. This suggests that this mechanism is likely to be involved in both acute and chronic allograft rejection. The local renal uPAR plays a pivotal role in the pathogenesis of I/R injury and allogenic transplant rejection. Specifically, uPAR deficiency protected kidney tissue from generation of ROS and, consecutively, from severe apoptosis in I/R injury and acute kidney allograft rejection [37]. It should be emphasized that αMUPA mice displayed the lowest basal renal expression of uPAR among the studied subgroups. Furthermore, the renal expression of uPAR increased following AKI in WT (males and females) and αMUPA males, but declined in αMUPA females. On the other hand, there have also been reports connecting the bone marrow to kidney damage involving the soluble urokinase plasminogen activator receptor (suPAR). suPAR is the circulating form of a three-domain membrane protein anchored by glycosylphosphatidylinositol [73,74,75,76,77]. This receptor is usually expressed on several cells at very low levels, like endothelial cells, podocytes, and in immunologically active cells, such as monocytes and lymphocytes, with induced expression [78,79,80]. Therefore, SuPAR levels are a strong predictor of progressive kidney function decline [73,81,82,83]. Long-term exposure to excessive suPAR levels specifically affects the kidneys through abnormal stimulation of podocyte-expressed alpha-vβ3 integrin, resulting in proteinuria [74,77,84]. An additional study found that suPAR in many cohorts was associated with subsequent AKI. The authors studied 4769 patients who were exposed to intra-arterial contrast material for coronary angiography prior to cardiac surgery, or who were critically ill; the authors obtained experimental proof that suPAR may be specifically involved in AKI pathogenesis through sensitizing proximal kidney tubules to injury by modulating cellular bioenergetics and increasing oxidative stress [85]. An additional aspect of uPA/uPAR system is its involvement in both innate and adaptive immune-mediated responses, as recently demonstrated [86]: specifically, the uPA/uPAR axis modulates the synthesis and presentation of antigen [87], lymphocyte activation [88], pro- and anti-inflammatory signal production [89], intracellular signaling pathway activation [90], cytotoxicity [91], cell adhesion [34], and migration [92,93,94], all of which are crucial phases throughout the immune response mediated by cells. Recently, a significant correlation was found between the uPA/uPAR system and the complementary system in the control of kidney immunological responses [95]. 

## 5. Conclusions

In summary: Kidneys of αMUPA mice, especially the females, are less susceptible to ischemic I/R injury compared to those of WT mice. This is due to nephroprotective actions mediated by direct indirect contribution of uPA via upregulation of leptin, eNOS, ACE2, and PGC1α along impaired inflammatory, fibrotic, and autophagy processes. The mechanisms by which leptin modulates gene expression following AKI requires further research; yet, these initial results reinforce leptin as a nephroprotective hormone. Nonetheless, we observed that uPA expression in the kidneys of αMUPA female mice was different, and even reduced, compared to kidneys of WT mice (Figure 3A). This emphasizes the indirect contribution of uPA in the resistance of the kidney as well as the heart, and not necessarily via direct effects.

## Figures and Tables

**Figure 1 cells-12-02497-f001:**
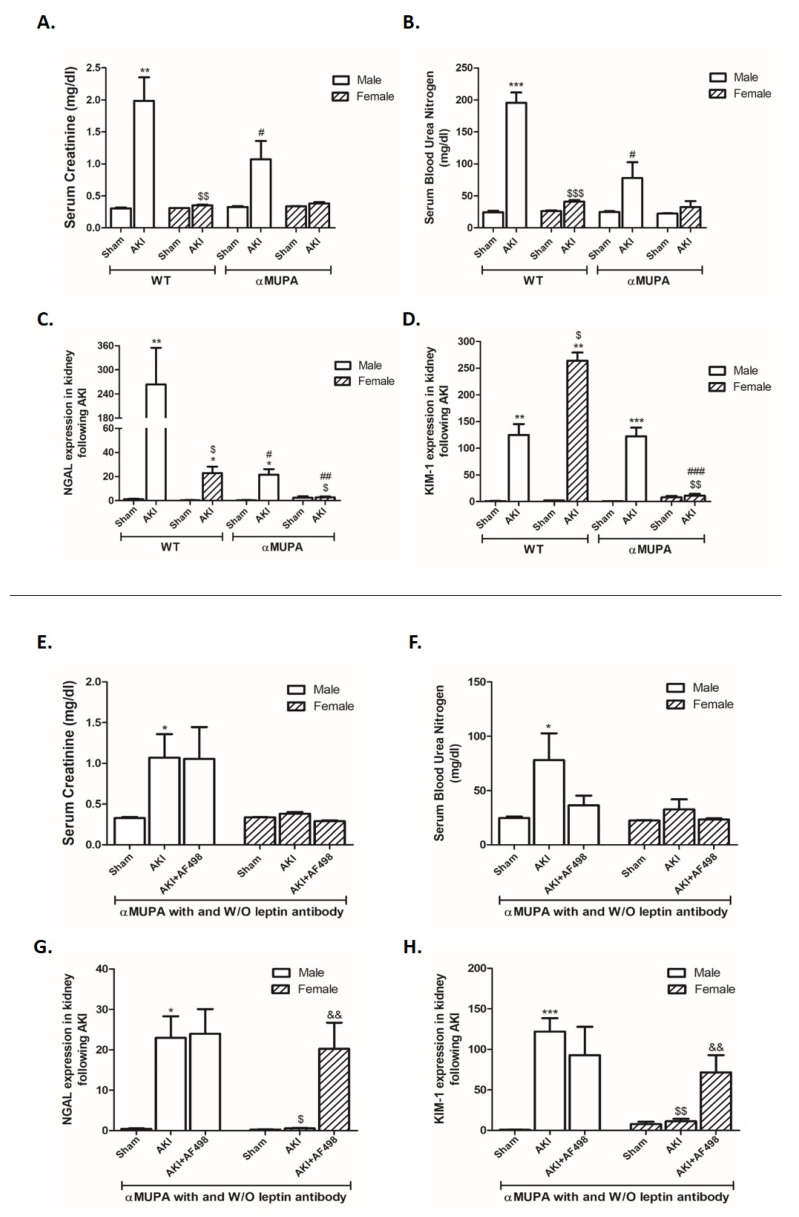
The impact of AKI on kidney function/renal injury in WT and αMUPA mice in the absence and presence of AF498 pretreatment. (**A**) Serum creatinine level; (**B**) blood urea nitrogen (BUN) level; (**C**) q-PCR of renal NGAL; (**D**) q-PCR of renal KIM-1; (**E**) serum creatinine level in αMUPA mice pretreated with AF498; (**F**) BUN level in αMUPA mice pretreated with AF498; (**G**) q-PCR of NGAL in αMUPA mice pretreated with AF498; (**H**) q-PCR of renal KIM-1 in αMUPA mice pretreated with AF498 (*, *p* < 0.05; **, *p* < 0.01; ***, *p* < 0.001—sham vs. AKI in the same group; $, *p* < 0.05; $$, *p* < 0.01; $$$, *p* < 0.001—male vs. female in the same mice strain; #, *p* < 0.05; ##, *p* < 0.01; ###, *p* < 0.001—WT vs. αMUPA that underwent similar procedure; &&, *p* < 0.01—before vs. after AF498 administration).

**Figure 2 cells-12-02497-f002:**
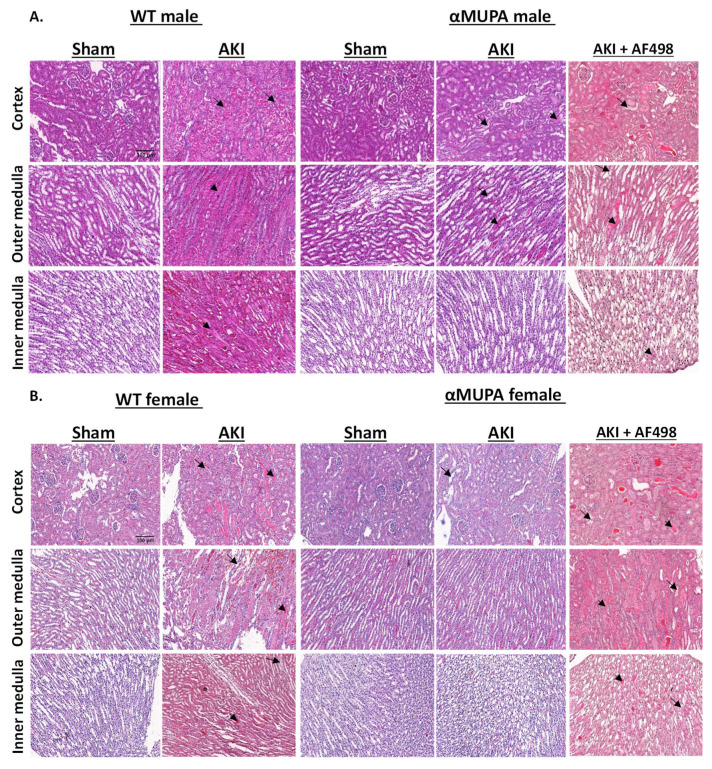
Effects of AKI on renal histology in WT and αMUPA in the absence and presence of AF498 pretreatment. (**A**) Representative cortical, outer medullary, and inner medullary histological sections of kidneys stained with hematoxylin and eosin taken from WT and αMUPA male mice: male sham WT mice (first column), male AKI WT mice (second column), male sham αMUPA mice (third column), male AKI αMUPA mice (fourth column), male AKI αMUPA mice pretreated with AF498 (fifth column). (**B**) The sections taken from kidneys of WT and αMUPA female mice. The long arrows indicate tubular collapse, loss of the brush border, and cellular detachment from tubular basement membranes; the short arrows indicate congestion. Images were taken at ×20 magnification; scale bar = 100 µm.

**Figure 3 cells-12-02497-f003:**
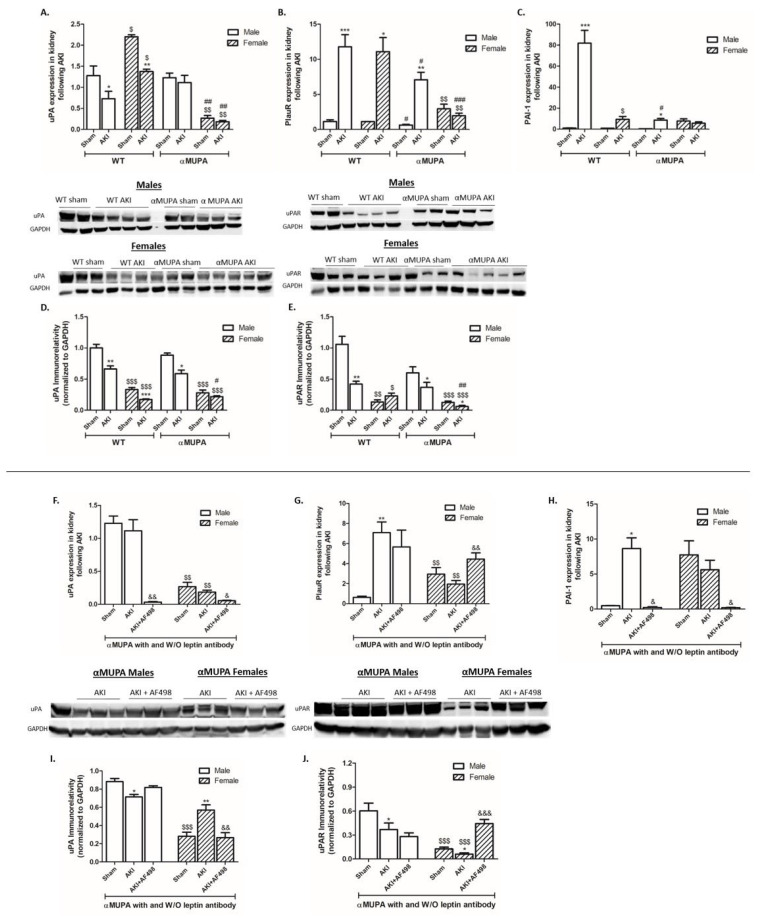
The impact of AKI renal expression and abundance of uPA and uPAR in the absence and presence of AF498 pretreatment. (**A**) Urokinase plasminogen activator (uPA) expression; (**B**) urokinase plasminogen receptor (PlauR/uPAR) expression; (**C**) plasminogen activator inhibitor 1 (PAI-1) expression; (**D**) immunoreactive levels of urokinase plasminogen activator (uPA); (**E**) immunoreactive levels of urokinase plasminogen receptor (PlauR /uPAR); (**F**) uPA expression following AF498; (**G**) PlauR /uPAR expression following AF498; (**H**) PAI-1 expression following AF498; (**I**) uPA abundance following AF498; (**J**) uPAR abundance following AF498 (*, *p* < 0.05; **, *p* < 0.01; ***, *p* < 0.001—sham vs. AKI in the same group; $, *p* < 0.05; $$, *p* < 0.01; $$$, *p* < 0.001—male vs. female in the same mice strain; #, *p* < 0.05; ##, *p* < 0.01; ###, *p* < 0.001—WT vs. αMUPA that underwent similar procedure; &, *p* < 0.05; &&, *p* < 0.01; &&&, *p* < 0.001—before vs. after AF498 administration).

**Figure 4 cells-12-02497-f004:**
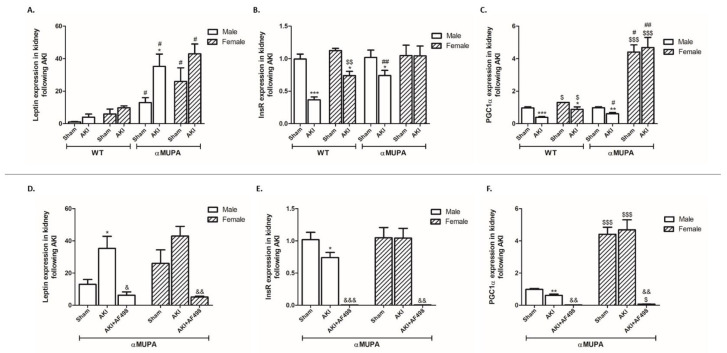
The impact of AKI on the expression of renal leptin, insulin receptor, and PGC1-α in WT and αMUPA mice in the presence and absence ofAF498 pretreatment. (**A**) Leptin; (**B**) insulin receptor (InsR); (**C**) PGC1α; (**D**) leptin following AF498; (**E**) InsR following AF498; (**F**) PGC1α following AF498 (*, *p* < 0.05; **, *p* < 0.01; ***, *p* < 0.001—sham vs. AKI in the same group; $, *p* < 0.05; $$, *p* < 0.01; $$$, *p* < 0.001—male vs. female in the same mice strain; #, *p* < 0.05; ##, *p* < 0.01—WT vs. αMUPA that underwent similar procedure; &, *p* < 0.05; &&, *p* < 0.01; &&&, *p* < 0.001—before vs. after AF498 administration).

**Figure 5 cells-12-02497-f005:**
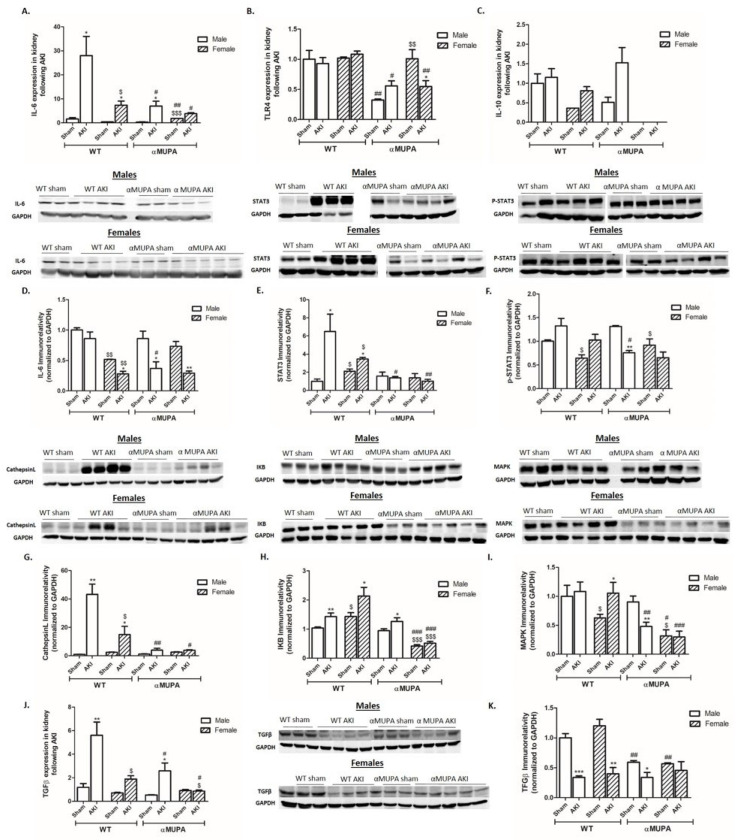
The impact of AKI on renal expression/abundance of inflammatory and fibrotic markers in the presence and absence ofAF498 pretreatment. (**A**) Expression of IL-6; (**B**) expression of Toll like receptor 4 (TLR4); (**C**) expression of IL-10; (**D**) immunoreactive levels of IL-6; (**E**) immunoreactive levels of STAT-3; (**F**) immunoreactive levels of p-STAT3; (**G**) immunoreactive levels of Cathepsin L; (**H**) immunoreactive levels of IKB; (**I**) immunoreactive levels of MAPK; (**J**) expression of TGF-β; (**K**) immunoreactive levels of TGF-β; (**L**) expression of IL-6 following AF498; (**M**) expression of TLR4 following AF498; (**N**) immunoreactive levels of STAT-3 following AF498; (**O**) immunoreactive levels of p-STAT3 following AF498; (**P**) immunoreactive levels of IKB following AF498; (**Q**) immunoreactive levels of MAPK following AF498; (**R**) expression of IL-10 following AF498; (**S**) expression of TGF-β following AF498 (*, *p* < 0.05; **, *p* < 0.01; ***, *p* < 0.001—sham vs. AKI in the same group; $, *p* < 0.05; $$, *p* < 0.01; $$$, *p* < 0.001—male vs. female in the same mice strain; #, *p* < 0.05; ##, *p* < 0.01; ###, *p* < 0.001—WT vs. αMUPA that underwent similar procedure; &, *p* < 0.05; &&, *p* < 0.01; &&&, *p* < 0.001—before vs. after AF498 administration).

**Figure 6 cells-12-02497-f006:**
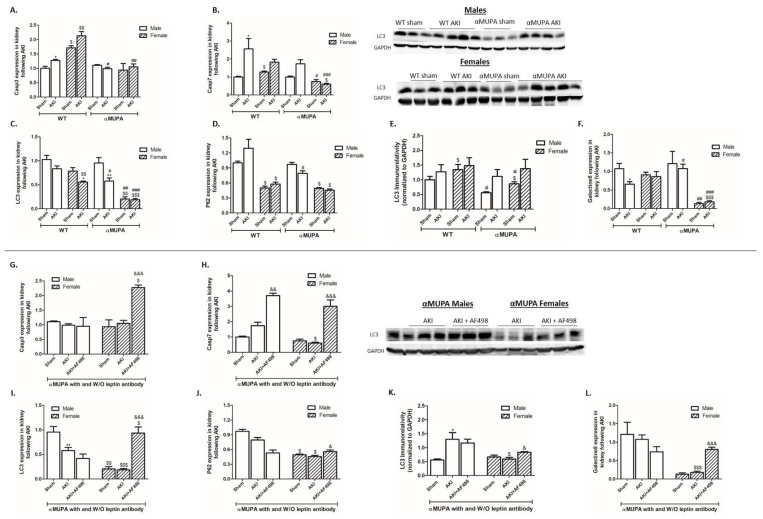
Impact of AKI on renal expression/abundance of apoptotic and autophagy markers in the presence and absence ofAF498 pretreatment: (**A**) expression of Caspase 3; (**B**) expression of Caspase 7; (**C**) expression of LC3; (**D**) expression of P62; (**E**) immunoreactive levels of LC3; (**F**) expression of Galectine 8; (**G**) expression of Caspase 3 following AF498; (**H**) expression of Caspase 7 following AF498; (**I**) expression of LC3 following AF498; (**J**) expression of P62 following AF498; (**K**) immunoreactive levels of LC3 following AF498; (**L**) expression of Galectine 8 following AF498 (*, *p* < 0.05; **, *p* < 0.01—sham vs. AKI in the same group; $, *p* < 0.05; $$, *p* < 0.01; $$$, *p* < 0.001—male vs. female in the same mice strain; #, *p* < 0.05; ##, *p* < 0.01; ###, *p* < 0.001—WT vs. αMUPA that underwent similar procedure; &, *p* < 0.05; &&, *p* < 0.01; &&&, *p* < 0.001—before vs. after AF498 administration).

**Figure 7 cells-12-02497-f007:**
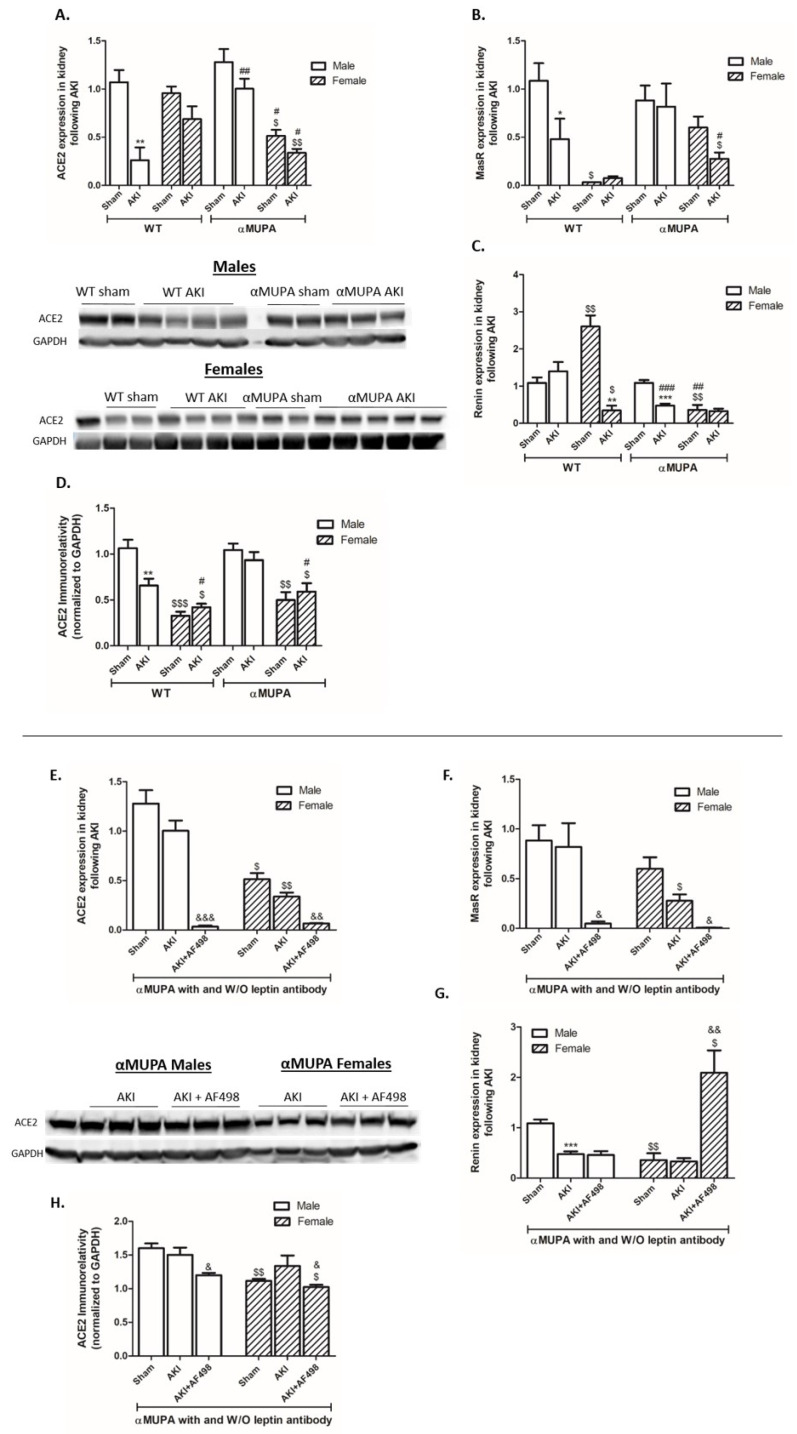
Effect of AKI on renal ACE2, MasR, and renin in the presence and absence ofAF498 pretreatment. (**A**) Expression of ACE2; (**B**) expression of Mas receptor; (**C**) expression of renin; (**D**) ACE2 immunoreactive levels amount; (**E**) expression of ACE2 following AF498; (**F**) expression of Mas receptor following AF498; (**G**) expression of renin following AF498; (**H**) ACE2 immunoreactive levels amount following AF498 (*, *p* < 0.05; **, *p* < 0.01; ***, *p* < 0.001—sham vs. AKI in the same group; $, *p* < 0.05; $$, *p* < 0.01; $$$, *p* < 0.001—male vs. female in the same mice strain; #, *p* < 0.05; ##, *p* < 0.01; ###, *p* < 0.001—WT vs. αMUPA that underwent similar procedure; &, *p* < 0.05; &&, *p* < 0.01; &&&, *p* < 0.001—before vs. after AF498 administration).

**Figure 8 cells-12-02497-f008:**
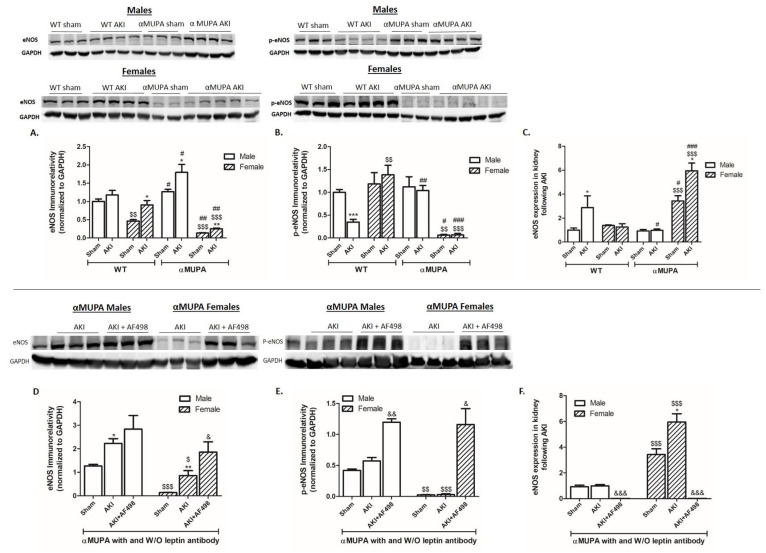
Effect of AKI on renal expression of eNOS and p-eNOS in the presence and absence ofAF498 pretreatment: (**A**) eNOS immunoreactivity; (**B**) p-eNOS immunoreactivity; (**C**) expression of eNOS; (**D**) eNOS immunoreactivity following AF498; (**E**) p-eNOS immunoreactivity following AF498; (**F**) expression of eNOS following AF498 (*, *p* < 0.05; **, *p* < 0.01; ***, *p* < 0.001—sham vs. AKI in the same group; $, *p* < 0.05; $$, *p* < 0.01; $$$, *p* < 0.001—male vs. female in the same mice strain; #, *p* < 0.05; ##, *p* < 0.01; ###, *p* < 0.001—WT vs. αMUPA that underwent similar procedure; &, *p* < 0.05; &&, *p* < 0.01; &&&, *p* < 0.001—before vs. after AF498 administration).

**Figure 9 cells-12-02497-f009:**
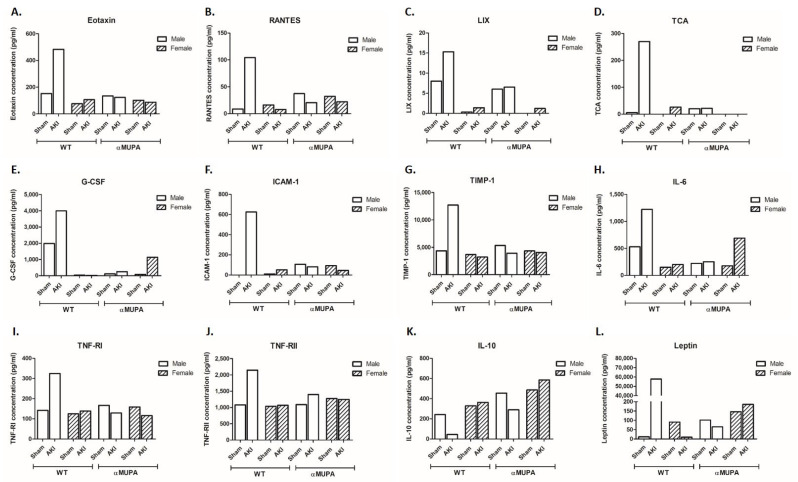
Serum concentration of chemokines and cytokines in WT and αMUPA mice (males and females) following AKI: (**A**) eotaxin; (**B**) RANTES; (**C**) LPS-induced CXC chemokine (LIX); (**D**) tricarboxylic acid (TCA); (**E**) granulocyte colony stimulating factor (G-CSF); (**F**) Intercellular Adhesion Molecule 1 (ICAM-1); (**G**) tissue inhibitor of metalloproteinases 1 (TIMP-1); (**H**) Interleukin 6 (IL-6); (**I**) tumor necrosis factor α receptor I (TNF-RI); (**J**) tumor necrosis factor α receptor II (TNF-RII); (**K**) Interleukin 10 (IL-10); (**L**) leptin.

**Table 1 cells-12-02497-t001:** Effects of AKI after 48 h on body and kidney weight in WT and αMUPA mice.

Group	Body Weight before Surgery (g)	Body Weight after Surgery (g)	% Change in BW	Kidney Weight (KW) (g)	Kidney/Body Weight Ratio (%)
WT male—sham	28.64 ± 0.77	27.57 ± 0.72	−3.7%	0.197 ± 0.007	0.71 ± 0.02
WT male—AKI	30.37 ± 0.82	26.94 ± 0.95 @	−11.3%	0.265 ± 0.01	0.99 ± 0.05 ***
WT female—sham	23.63 ± 0.3	22.95 ± 1.1	−2.8%	0.13 ± 0.007	0.59 ± 0.0005 $
WT female—AKI	22.81 ± 1.04	20.96 ± 0.63	−8%	0.17 ± 0.008	0.81 ± 0.02 **,$
αMUPA male—sham	22.84 ± 0.75	23.63 ± 1.4	+3.4%	0.16 ± 0.007	0.71 ± 0.01
αMUPA male—AKI	23.49 ± 0.42	21.91 ± 0.28 @@	−6.7%	0.174 ± 0.005	0.79 ± 0.02 *,#
αMUPA male—AKI + AF498	28.73 ± 0.58	26.26 ± 0.69	−8.6%	0.254 ± 0.02	0.96 ± 0.05
αMUPA female—sham	20.42 ± 1	21.56 ± 0.13	+5.5%	0.113 ± 0.003	0.52 ± 0.02 $$
αMUPA female—AKI	20.75 ± 0.65	19.21 ± 0.47	−7.4%	0.114 ± 0.003	0.59 ± 0.02 $$$,###
αMUPA male—AKI + AF498	20.08 ± 1.03 $$	17.74 ± 0.67 $$	−11.25%	0.147 ± 0.02	0.82 ± 0.09

Body and kidney weight of males and females mice, before and after AKI, in the presence or absence of AF498; @, *p* < 0.05; @@, *p* < 0.01—body weight before vs. body weight after; *, *p* < 0.05; **, *p* < 0.01; ***, *p* < 0.001—sham vs. AKI in the same group; $, *p* < 0.05; $$, *p* < 0.01; $$$, *p* < 0.001—male vs. female in the same mice strain; #, *p* < 0.05; ###, *p* < 0.001—WT vs. αMUPA that underwent similar procedure.

## Data Availability

All data are available from the corresponding author upon a reasonable request.

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
