# Peer review of "The Protective Pathways Activated in Kidneys of αMUPA Transgenic Mice Following Ischemia\Reperfusion-Induced Acute Kidney Injury"

_cells, 2023, doi:10.3390/cells12202497_

Round 1
Reviewer 1 Report
Thank you for the opportunity to review this work by Abd.Elkhaleq et al. The authors have performed an interesting study using a mouse model known to have extended life and resistance to ischemia to probe the protective mechanisms of leptin in renal i/r AKI. In general, the experimental approach is sound and the results are exciting. My two major critiques are thus:
1) The authors are to be commended for the inclusion of both sexes in their study, which strengthens their overall approach significantly, as well as their summary of the known sex biases in i/r AKI. However, how do you interpret the relative protective effects of MUPA vs female sex? In general, this experimental approach using a consistent clamp time between the sexes is easier and straightforward, however it would perhaps be of greater interest to use a model in which the degree of AKI is matched between the sexes (longer clamp time in females compared to males, (see PMID: 35022484). While beyond the scope of this current work, it may merit consideration within the discussion that the matched ischemia duration and therefore differential AKI between the sexes is a limitation of the current study.
2) The results are at times difficult to read/interpret secondary to the table used and/or symbols rather than using figures with typical significance standards (*, **, ***, etc) and connecting bars to distinguish the groups. For this reason, it took quite a bit of time to try to read and interpret the manuscript, and I would strongly encourage the authors to rework the figures so that it is not difficult to readers to interpret.
Otherwise, please see my point-by-point comments below that are meant to be constructive and will hopefully strengthen the final product.
Abstract:
· Alpha-MUPA needs to be defined before the acronym is used, within the abstract.
· The type of i/r AKI should be stated – bilateral (or unilateral) and include clamp time.
· ‘Interestingly’ is rather editorial. Would merely state the results. ‘Interestingly’ is used quite a bit throughout the results section as well, would avoid the editorialization.
· Would not state that kidney function was ‘measured’ since GFR was not measured, only biomarkers to estimate kidney function were measured prior to sacrifice.
Intro:
· Define CKD, WT, MI, TNF-, IL-, MUPA and LPS acronyms before first use.
· Define MUPA acronym before first use. What alpha-MUPA mice are, what MUPA stands for, has not been at all described in the introduction. It is assumed that the reader is familiar with this strain.
Methods:
· What age were the mice at the time of AKI?
· The surgical approach should be clarified (it sounds ventral, but please state as such in the methods).
· How was the “impact” of the surgery evaluated at 48 hours? Was there an exclusion cutoff for certain biomarkers (i.e. BUN or SCr) for the AKI cohorts? Was it the same for both sexes?
· Were results analyzed unpooled with regards to sex?
· Was there no fluid repletion after surgery?
· What was the mortality of the WT and alpha-MUPA mice after AKI and sham procedures?
Results/Discussion:
· How was KW assessed pre and post AKI? Or are you just comparing KW in AKI cohorts to their sham counterparts? If the latter, would clarify the results section accordingly. As written, it sounds as though you measured KW pre AKI and post AKI in each mouse.
· It may be more appropriate and helpful to report the %change in BW rather than the actual decrease in weight.
· What functional role/importance does KW play? Why measure it? Using changes in KW to assess the efficacy of leptin/leptin-ab therapy needs to be rationalized. Otherwise, biomarkers of kidney function and/or histology should be used to assess this mechanism.
· The table should be presented as a figure so readers can interpret the results, particularly in light of the numerous comparisons (WT vs MUPA, sham vs AKI, male vs female). Furthermore, the various symbols are difficult to follow, making this table unfortunately unintelligible.
· Figures: significance between the groups is similarly difficult to follow. Would recommend using the *, ** system amongst the same groups, and then showing horizontal bars with relative significance connecting various groups.
· Histology pictures would benefit from increased zoom. Scale bar should also be included. It would also be helpful to guide readers through the differences in the results, either with arrows showing inflammatory cells, or some other means.
· Numerous acronyms are used without prior definition (i.e. IKB, TLR-4, MAPK)
· Figure 5 is too small to be legible, even after zooming in considerably (200% is still too small). Please ensure that print article readers will be able to read/interpret the findings. Figure 6 is also rather small, but not as bad as Figure 5.
· Throughout, I do not think it is appropriate to compare the female groups to the males given the known sex biases in i/r AKI. Only in a model with a matched degree of AKI in males and females does this comparison make sense.
Minor comments:
· Would avoid the use of boldtype which has been smattered throughout. (For example page 19, lines 487-488). There are also random lines using italics and/or different font size, perhaps secondary to copy-editing issues.

· There are minor syntax/grammatical errors throughout the introduction that may benefit from formal grammatical review.
· A few typos in the discussion (ntreated, bassline).
Author Response
Thank you for the opportunity to review this work by Abd. Alkhaleq et al. The authors have performed an interesting study using a mouse model known to have extended life and resistance to ischemia to probe the protective mechanisms of leptin in renal i/r AKI. In general, the experimental approach is sound and the results are exciting. My two major critiques are thus:
- The authors are to be commended for the inclusion of both sexes in their study, which strengthens their overall approach significantly, as well as their summary of the known sex biases in i/r AKI. However, how do you interpret the relative protective effects of MUPA vs female sex? In general, this experimental approach using a consistent clamp time between the sexes is easier and straightforward, however it would perhaps be of greater interest to use a model in which the degree of AKI is matched between the sexes (longer clamp time in females compared to males, (see PMID: 35022484). While beyond the scope of this current work, it may merit consideration within the discussion that the matched ischemia duration and therefore differential AKI between the sexes is a limitation of the current study.
- The results are at times difficult to read/interpret secondary to the table used and/or symbols rather than using figures with typical significance standards (*, **, ***, etc) and connecting bars to distinguish the groups. For this reason, it took quite a bit of time to try to read and interpret the manuscript, and I would strongly encourage the authors to rework the figures so that it is not difficult to readers to interpret.
Otherwise, please see my point-by-point comments below that are meant to be constructive and will hopefully strengthen the final product.
Abstract:
- Alpha-MUPA needs to be defined before the acronym is used, within the abstract.
Response: αMUPA-alpha murine urokinase-type plasminogen activator transgenic mice, is now defined (see p.2).
- The type of i/r AKI should be stated – bilateral (or unilateral) and include clamp time.
Response: The type of I/R induced AKI is now specified, specifically a 30 min bilateral renal artery occlusion was applied to induce ischemic AKI (see p.2).
- ‘Interestingly’ is rather editorial. Would merely state the results. ‘Interestingly’ is used quite a bit throughout the results section as well, would avoid the editorialization.
Response: Thank you for the note. The word “Interestingly” was changed throughout the text in order to avoid bias attitude. - Would not state that kidney function was ‘measured’ since GFR was not measured, only biomarkers to estimate kidney function were measured prior to sacrifice.
Response: Thank you for the note. We avoided using kidney function, as we did not measure creatinine or inulin clearance. We used “biomarkers of kidney function/injury”.
Intro:
- Define CKD, WT, MI, TNF-, IL-, MUPA and LPS acronyms before first use.
Response: All the abbreviations are fully defined in the revised version. - Define MUPA acronym before first use. What alpha-MUPA mice are, what MUPA stands for, has not been at all described in the introduction. It is assumed that the reader is familiar with this strain.
Response: A short paragraph referring to alpha-MUPA mice was added to the introduction (see p.5).
Methods:
- What age were the mice at the time of AKI?
Response: 12 weeks (see methods p.6). - The surgical approach should be clarified (it sounds ventral, but please state as such in the methods).
Response: A more detailed description of the surgical procedure of AKI induction was added in the revised version (see methods p.6). - How was the “impact” of the surgery evaluated at 48 hours? Was there an exclusion cutoff for certain biomarkers (i.e. BUN or SCr) for the AKI cohorts? Was it the same for both sexes?
Response: We applied the same I/R time for all the experimental groups (males and females) except the sham operated- controls. As one of the aims of the current study was to compare between males and females we measured biomarkers of kidney function and renal injury along histological analysis without being limited to certain cutoff values. The main parameter that we adhere to is the I/R duration which was identical in all groups and comparison of the basal values and control groups. - Were results analyzed unpooled with regards to sex?
Response: NO, we expect that the resistance of females to AKI will be offset by the sensitivity of males to the disease. - Was there no fluid repletion after surgery?
Response: NO, the animals were recovered quickly as the anesthesia was not long lasting and water consumption was resumed relatively swiftly. - What was the mortality of the WT and alpha-MUPA mice after AKI and sham procedures?
Response: The mortality rate after AKI was a twice-in WT (around 30%) as compare to alpha-MUPA mice (around 15%). It should be emphasized, that the mortality rate in females was lower than that of males.
Results/Discussion:
- How was KW assessed pre and post AKI? Or are you just comparing KW in AKI cohorts to their sham counterparts? If the latter, would clarify the results section accordingly. As written, it sounds as though you measured KW pre AKI and post AKI in each mouse.
Response: In table 1, kidney weight/body weight after 48 hours from AKI of the various experimental groups is presented. That means we just compared KW in AKI cohorts to their sham counterparts. It is obvious we could not compare kidney weight before and after AKI in the same animal. Clarification was added (see results p.10 and updated table 1). - It may be more appropriate and helpful to report the %change in BW rather than the actual decrease in weight.
Response: Thank you for the note, we modified table 1 so the percentage change in BW is now included (see Table 1). - What functional role/importance does KW play? Why measure it? Using changes in KW to assess the efficacy of leptin/leptin-ab therapy needs to be rationalized. Otherwise, biomarkers of kidney function and/or histology should be used to assess this mechanism.
Response: The increase in KW reflects the development of renal edema and inflammatory response. The effect of leptin in reducing KW indicates beneficial action of this hormone. In line with this concept when leptin blocked by AF498 the KW/BW was aggravated. In addition, leptin blocking worsen biomarkers of renal injury and the adverse histological alterations. - The table should be presented as a figure so readers can interpret the results, particularly in light of the numerous comparisons (WT vs MUPA, sham vs AKI, male vs female). Furthermore, the various symbols are difficult to follow, making this table unfortunately unintelligible.
Response: Usually BW and KW are presented in table rather than in figure, so the reader can notice the precise values as the changes sometimes are minor. Since we have multiple comparison we have to use different symbols. - Figures: significance between the groups is similarly difficult to follow. Would recommend using the *, ** system amongst the same groups, and then showing horizontal bars with relative significance connecting various groups.
Response: We understand the difficulties in following up the applied symbols between the various groups. However, since we have 8 groups, applying horizontal bars will create crowded figures which already overwhelming. We use * to compare between sham and AKI regardless the gender and the strain, $ to compare between males and females in the same strain that underwent the same procedure, and # to compare between WT and MUPA mice of the same gender that underwent the same surgical procedure. We applied this clue in all the relevant legends. - Histology pictures would benefit from increased zoom. Scale bar should also be included. It would also be helpful to guide readers through the differences in the results, either with arrows showing inflammatory cells, or some other means.
Response: Arrows are added to indicate the cellular detachment (long arrows) and congestion (short arrows), in addition the scale bar is added (see Fig 2.) - Numerous acronyms are used without prior definition (i.e. IKB, TLR-4, MAPK)
Response: Thank you for the notice, all the acronyms are now defined. - Figure 5 is too small to be legible, even after zooming in considerably (200% is still too small). Please ensure that print article readers will be able to read/interpret the findings. Figure 6 is also rather small, but not as bad as Figure 5.
Response: Thank you for the notice. In order to overcome this issue we divided Figure 5 into two panels, panel 1 include the results without AF498 (A-K) and panel 2 include the results with AF498 (L-S). - Throughout, I do not think it is appropriate to compare the female groups to the males given the known sex biases in i/r AKI. Only in a model with a matched degree of AKI in males and females does this comparison make sense.
Response: Although we understand the reviewer concern we chose the same I/R period (30 min) rather than using matched degree of AKI. The sense behind our approach is to compare the vulnerability of αMUPA to AKI as compared to their WT in one hand, and between males and females subjected to the same renal ischemic insult on the other. In other words we wanted to establish this phenomenon before applying match degree of AKI which could be subject of future study. If we were to apply matched degree of the AKI, then we may miss the differences between the strains and the genders of αMUPA and their WT (FVB/N).
Minor comments:
- Would avoid the use of boldtype which has been smattered throughout. (For example page 19, lines 487-488). There are also random lines using italics and/or different font size, perhaps secondary to copy-editing issues.
Response: Done. - There are minor syntax/grammatical errors throughout the introduction that may benefit from formal grammatical review.
Response: The whole manuscript and grammar editing. - A few typos in the discussion (untreated, bassline).
Response: Done

Reviewer 2 Report
In the present study, the effects of gender, αMUPA, and leptin on the pathomechanism of AKI were examined. The question is interesting and the text is relatively easy to follow.
However, I find it regrettable that the complex question was investigated using a single animal model, it would be nice to see if the measurements carried out on an independent model (e.g. in vitro) lead to similar results. This would have helped to understand also the more exact relationships between the examined factors.
In general my main problems are as follows:
A broader interpretation of the applied transgenic animals would be necessary.
Unfortunately, only the αMUPA mice were treated with the leptin neutralizing antibody (AF498) (where the effect of αMUPA is also present), therefore it was not possible to estimate the effect of leptin itself on the pathomechanism of AKI. Overall, due to the many factors examined, the work does not have a clear logical thread. This is a bit subjective, but I would prefer a project investigating only the effect of leptin (groups: sham, sham+AF498, AKI, AKI+AF498).
Other suggestions:
Abstract:
What does αMUPA transgenic mice cover? Are these mice overexpress uPA?
I would avoid using expressions like "high expression" (line 26)
Introduction:
During the project, the relationship between I/R damage, gender, αMUPA and leptin was investigated (these are the variables of the model), the other measured parameters are only used to describe the effect of these variables. Therefore, the sections between lines 42 and 55 and lines 58/59 and 75 is not necessary.
The logical link between gender, uPA and leptin is missing. How does each influence the other? What is the role of uPA in AKI?
Line 84 according to which "..but not uPA expression plays a pivotal role in the pathogenesis of a mice I/R AKI model...." is a bit strange, while using uPA transgenic mice in the present study.
Methods:
I recommend to create separate sections for each molecular biology methods, as follows:
2.3 Histopathology
2.4 Real-time PCR
2.5 Western blot
A description of the primers used during PCRs is missing.
Results:
During the experiment, the following groups were examined:
WT sham, male
WT AKI, male
WT sham, female
WT AKI, female
αMUPA sham, male
αMUPA AKI, male
αMUPA AKI+AF498 male
αMUPA sham, female
αMUPA AKI, female
αMUPA AKI+AF498 female
The results from the above groups are sometimes displayed redundantly (eg, Figure 1). In Figure 1, the results of the two mice treated with AF498 could be inserted into diagram A, so that diagram E could be removed from the manuscript. The same is true for the other diagrams in Figure 1, as well. The situation is similar in case of Figure 4, 5, 6, and partly in Figure 7.
The resolution of the histological images is not sufficient for their analysis by the reader.
The figure captions do not follow the figures (e.g. what does it statistically mean if there are two or more stars/dollar signes, etc. in the figures?).
The Y-axis of the figures displaying Western blot measurements somewhere indicate expression and somewhere immunoreactivity, I recomend the unification of the captation of the axises.
In general it helps the understanding if we talk about the mRNA expression of the given molecule and the amount of protein in the text.
In some places, there is an interpretation of the results which is unnecessary in the section of results (e.g. lines 226-227).
Discussion:
Since many diseases, in fact the majority of them, are multifactorial and yet have pharmacological treatment, I cannot agree with the 2. sentence of the discussion (Since AKI is a multifactorial disease, there is no effective pharmacological therapy that prevents the progression of the disease).
A broader interpretation of the applied transgenic animals would be necessary. In my reading, when creating the transgenic mice Miskin et al. inserted the coding sequence of the uPA gene just downstream of αA-crystallin promoter and this gene construct was microinjected into the fertilised mouse eggs (ref 41). Consequently, since the αA-crystallin promoter is exclusively active in the eye lens, the introduced uPA gene must be expressed only in the eye lens of the resulting transgenic mice, as it is also described by Miskin et al. If this is the case, what explains that the uPA expression in the kidneys of female transgenic mice is different, moreover reduced, compared to that of WT animals (Figure 3/A)?
It must be also note that the level of the uPA protein (which may carry the biological effect) does not differ between WT and transgene sham animals. If this is so, what proves that the mice were transgenic?
Minor comment:
The abbreviations sometimes are not resolved.
Although there are inaccuracies in the text, the manuscript is generally easy to understand.
Author Response
REVIEWER II
Comments and Suggestions for Authors
In the present study, the effects of gender, αMUPA, and leptin on the pathomechanism of AKI were examined. The question is interesting and the text is relatively easy to follow.
However, I find it regrettable that the complex question was investigated using a single animal model, it would be nice to see if the measurements carried out on an independent model (e.g. in vitro) lead to similar results. This would have helped to understand also the more exact relationships between the examined factors.
In general my main problems are as follows:
A broader interpretation of the applied transgenic animals would be necessary.
Unfortunately, only the αMUPA mice were treated with the leptin neutralizing antibody (AF498) (where the effect of αMUPA is also present), therefore it was not possible to estimate the effect of leptin itself on the pathomechanism of AKI. Overall, due to the many factors examined, the work does not have a clear logical thread. This is a bit subjective, but I would prefer a project investigating only the effect of leptin (groups: sham, sham+AF498, AKI, AKI+AF498).
Response: Thank you for valuable note. Indeed, administration of leptin neutralizing antibody should have been perform on both WT and αMUPA. We were aware to this matter, yet we treated only αMUPA mice with AF498. This was done due to financial limitation as these antibodies are very expensive and injection to so many mice would require substantial amount of money. In addition, since we apply αMUPA mice that were studied at the cardiac level and were found to be resistant to I/R induced MI as compared to WT mice [1]. In αMUPA mice but not in WT animals, administration of AF498 abolished their resistance to I/R induced MI [ PLoS One, 2015. 10(12): p. e0144593.]. Therefore, we were encourage by these results and applied the administration of leptin neutralizing antibody only for αMUPA mice.
Other suggestions:
Abstract:
What does αMUPA transgenic mice cover? Are these mice overexpress uPA?
Response: Thank you for the note. A more detailed description is now included in the abstract.
I would avoid using expressions like "high expression" (line 26)
Response: Thank you for the note. Corrected (see abstract).
Introduction:
During the project, the relationship between I/R damage, gender, αMUPA and leptin was investigated (these are the variables of the model), the other measured parameters are only used to describe the effect of these variables. Therefore, the sections between lines 42 and 55 and lines 58/59 and 75 is not necessary.
Response: Thank you for the note. We removed many of the suggested sections, however we rather to leave those refer to the NO involvement in the pathogenesis of the AKI.
The logical link between gender, uPA and leptin is missing. How does each influence the other? What is the role of uPA in AKI?
Response: Thank you for the note. In the introduction (page 4, 2nd -5) we wrote in details why we think that uPA could be involved in the pathogenesis of AKI and the availability of αMUPA mice may serve as an ideal platform to address this question.
We rephrase the last paragraph in the introduction to focus on gender, uPA and leptin relationships.
Line 84 according to which "..but not uPA expression plays a pivotal role in the pathogenesis of a mice I/R AKI model...." is a bit strange, while using uPA transgenic mice in the present study.
Response: Thank you for the note. We regret for this mistake and following statement: “Local renal uPAR, but not uPA expression plays a pivotal…. “ was changed to: “Local renal uPAR expression plays a pivotal….” (Page 4).
Methods:
I recommend to create separate sections for each molecular biology methods, as follows:
2.3 Histopathology
2.4 Real-time PCR
2.5 Western blot
A description of the primers used during PCRs is missing.
Response: Thank you for the notice. The edition is done and all utilized primers are now listed.
Results:
During the experiment, the following groups were examined:
WT sham, male
WT AKI, male
WT sham, female
WT AKI, female
αMUPA sham, male
αMUPA AKI, male
αMUPA AKI+AF498 male
αMUPA sham, female
αMUPA AKI, female
αMUPA AKI+AF498 female
The results from the above groups are sometimes displayed redundantly (eg, Figure 1). In Figure 1, the results of the two mice treated with AF498 could be inserted into diagram A, so that diagram E could be removed from the manuscript. The same is true for the other diagrams in Figure 1, as well. The situation is similar in case of Figure 4, 5, 6, and partly in Figure 7.
Response: Although we understand the sense behind inserting the AF498 treated mice with untreated animals, however the graphs are already crowded and the additional of 2 groups to the existing one will further complicate the presentation. Our aim from presenting AF498 treated and untreated αMUPA mice in separate group is to emphasize the impact of anti-leptin neutralizing Abs on the severity of AKI in these animals without diluting it will all groups which some of it is irrelevant.
The resolution of the histological images is not sufficient for their analysis by the reader.
Response: The resolution of the representative renal H&E pictures is high, however since the graph includes so many subgroups and different regions of the kidney, the size of the picture is small. We include PPT picture with resolution.
The figure captions do not follow the figures (e.g. what does it statistically mean if there are two or more stars/dollar signes, etc. in the figures?).
Response: Thank you for the note. We added appropriate symbols to reflect the range of statistical significance: *, P<0.05, **, P<0.01, ***, P<0.001; …….
The Y-axis of the figures displaying Western blot measurements somewhere indicate expression and somewhere immunoreactivity, I recomend the unification of the captation of the axises.
In general it helps the understanding if we talk about the mRNA expression of the given molecule and the amount of protein in the text.
Response: Done.
In some places, there is an interpretation of the results which is unnecessary in the section of results (e.g. lines 226-227).
Response: Done.
Discussion:
Since many diseases, in fact the majority of them, are multifactorial and yet have pharmacological treatment, I cannot agree with the 2. sentence of the discussion (Since AKI is a multifactorial disease, there is no effective pharmacological therapy that prevents the progression of the disease).
Response: Removed
A broader interpretation of the applied transgenic animals would be necessary. In my reading, when creating the transgenic mice Miskin et al. inserted the coding sequence of the uPA gene just downstream of αA-crystallin promoter and this gene construct was microinjected into the fertilised mouse eggs (ref 41). Consequently, since the αA-crystallin promoter is exclusively active in the eye lens, the introduced uPA gene must be expressed only in the eye lens of the resulting transgenic mice, as it is also described by Miskin et al. If this is the case, what explains that the uPA expression in the kidneys of female transgenic mice is different, moreover reduced, compared to that of WT animals (Figure 3/A)?
Response: You are right. Indeed Dr. Miskin reported that αMUPA mice supposed to exclusively express the uPA in the eye lens. However, analysis of the body organs for uPA revealed that it is also over-expressed in the brain which affect their appetite and food intake, leptin levels and subsequently may affect the behavior of other organs [2]. Moreover, uPA is expressed under baseline in various body organs including the kidney. Therefore, the utilization of these mice was driven by the observation that these animals were resistant to cardiac I/R damage [1]. The observation that uPA expression in the kidneys of female transgenic mice is different, and even reduced, compared to that of WT animals (Figure 3/A) emphasizes also the indirect contribution of uPA in the behavior of vital organs such as the heart and the kidney, not necessary via direct effects (we added this finding to the limitation of the study (Discussion, page 25).
It must be also note that the level of the uPA protein (which may carry the biological effect) does not differ between WT and transgene sham animals. If this is so, what proves that the mice were transgenic?
Response: Your statement is well taken. However, the observation that renal uPA immunoreactivity does not differ between WT and transgene sham animals support an indirect contribution of uPA in the resistance to I/R, via leptin related axis.
Minor comment:
The abbreviations sometimes are not resolved.
Response: Done
Comments on the Quality of English Language
Although there are inaccuracies in the text, the manuscript is generally easy to understand.
Response: The whole MS underwent English and grammar editing by English speaking professional.
References:
- Levy, E., et al., Long-Lived alphaMUPA Mice Show Attenuation of Cardiac Aging and Leptin-Dependent Cardioprotection. PLoS One, 2015. 10(12): p. e0144593.
- Miskin, R. and T. Masos, Transgenic mice overexpressing urokinase-type plasminogen activator in the brain exhibit reduced food consumption, body weight and size, and increased longevity. J Gerontol A Biol Sci Med Sci, 1997. 52(2): p. B118-24.

Round 2
Reviewer 2 Report
Thank you for the answers, they helped to understand the aim of the study, however important questions still remain open. Also thank for your answers, most of them is acceptable, however there are some still open questions. Please see my comments below.
I understand that the heart was protected against I/R injury in the aMUPA transgenic animal. I can accept that the overexpression of uPA also means protection in the kidney. I can also accept that the overexpression of uPA can only be observed in the eye and the brain.
Although it was not easy, finally I undertood that you probably suggest that the increased expression of uPA in the brain may affects the leptin level of the animals, which in turn affects the kidneys.
If it was the main question, it would be good to make it obvious in the manuscript. It would also be good to present literary data supporting the hypothesis.
The proposed hypothesis is interesting, however it would be good to see that the expression of uPA in the brain is really increased and that it really affects the synthesis of leptin. How does the serum leptin level of mice changing?
How do you explain the increased leptin production of the kidneys of the transgenic mice, does it affect the leptin level of the whole body, or does it affect the serum leptin level?
It is a separate question but how to explain the differences measured in the uPA expression/level in the kidney. Is it not possible that the effect of gender differences is manifested in the results?
not relevant
Author Response
Thank you for the answers, they helped to understand the aim of the study, however important questions still remain open. Also thank for your answers, most of them is acceptable, however there are some still open questions. Please see my comments below.
I understand that the heart was protected against I/R injury in the aMUPA transgenic animal. I can accept that the overexpression of uPA also means protection in the kidney. I can also accept that the overexpression of uPA can only be observed in the eye and the brain.
Comment 1- Although it was not easy, finally I understood that you probably suggest that the increased expression of uPA in the brain may affects the leptin level of the animals, which in turn affects the kidneys.
Comment 2- If it was the main question, it would be good to make it obvious in the manuscript. It would also be good to present literary data supporting the hypothesis.
Comment 3- The proposed hypothesis is interesting; however, it would be good to see that the expression of uPA in the brain is really increased and that it really affects the synthesis of leptin. How does the serum leptin level of mice changing?
Response to comments 1-3:
Thank you for the notes. We inserted into the introduction the following paragraph: “The alterations seen in αMUPA mice are turned on by uPA over-expression in the trigeminal nucleus in the brain. The trigeminal nucleus connected to the dorsal motor of the vagus and nucleus tractus solitarius, two regions of the brain that control appetite, this connection may help enervate adipose tissue through the autonomic nervous system, leading to an increase in the secretion of the adipokine hormone leptin. Leptin is a 16 Kd adipokine hormone produced by adipocytes, which controls the food intake and energy consumption [45, 46]. Due to elevated leptin levels in the serum, αMUPA mice have a balanced and resistant to obesity. Pro-opiomelanocortin (POMC) and cocaineand amphetamine-regulated transcript (CART) are expressed by anorexigenic neurons in the hypothalamus arcuate nucleus in response to leptin, a satiety signal produced by adipose tissue [47, 48]. uPA mRNA has also been detected in neuronal cells of the hypothalamus paraventricular nucleus (PVN), an area linked to eating behavior, in the αMUPA brain. The uPA mRNA could not be found at this site normally [43]” (See p.5), which actually addresses all your queries regarding the correlation between uPA and leptin. Previous studies demonstrated that αMUPA female and to a lesser extent male mice exhibit high levels of circulatory leptin as compared to their WT animals (PLoS One. 2017;12(11):e0188658; PLoS One, 2015. 10(12): p. e0144593).
Comment 4: How do you explain the increased leptin production of the kidneys of the transgenic mice, does it affect the leptin level of the whole body, or does it affect the serum leptin level?
Response: Unfortunately, the mechanisms underlying the increased expression of leptin in the renal tissue are not known. However, stressful stimuli such as ischemic and inflammatory responses may provoke local leptin production in various organs besides the adipose tissue. This notion is supported by previous studies that showed enhanced production of leptin in cardiac tissue in response to ischemic (https://doi.org/10.1016/j.yexcr.2020.112373) and inflammatory (10.1016/j.yexcr.2021.112647) stress. The major source of leptin is the adipose tissue, however we do not exclude that upregulation of leptin production by the kidney contributes to the elevated level of circulating leptin. It should be emphasized that renal sympathetic nerves innervate the vasculature and nephrons, where they play a central role in the regulation of renal hemodynamics, tubular function and probably local leptin expression (Physiol Rev. (1997) 77:75–197., 94; Clin Exp Pharmacol Physiol. (2004) 31:380–6). We discussed this possibility in the revised version of the MS (Discussion, page 21).
Comment 5: It is a separate question but how to explain the differences measured in the uPA expression/level in the kidney. Is it not possible that the effect of gender differences is manifested in the results?
Response: We took additional look at Figure 3 (Panel D) which depicts immunoreactive levels of uPA in the kidney of both αMUPA mice (males and females) and their wild type. It is obvious that basal renal uPA immunoreactivity is comparable in both subgroups. Moreover, this pattern persisted even after AKI, suggesting that basal renal uPA likely does not contribute to the gender differences in the results.